# *LSM12-EPAC1* defines a neuroprotective pathway that sustains the nucleocytoplasmic RAN gradient

**Jongbo Lee**[1], **Jumin Park**[1], **Ji-hyung Kim**[1], **Giwook Lee**[1], **Tae-Eun Park**[1], **Ki-Jun Yoon**[2], **Yoon Ki Kim**[3,4], **Chunghun Lim**[1]*

**1** School of Life Sciences, Ulsan National Institute of Science and Technology, Ulsan, Republic of Korea, **2** Department of Biological Sciences, Korea Advanced Institute of Science and Technology, Daejeon, Republic of Korea, **3** Creative Research Initiatives Center for Molecular Biology of Translation, Korea University, Seoul, Republic of Korea, **4** Division of Life Sciences, Korea University, Seoul, Republic of Korea

* clim@unist.ac.kr

**Data Availability Statement:** All relevant data are within the paper and its Supporting Information files. Raw and processed data from RNA

## Abstract

Nucleocytoplasmic transport (NCT) defects have been implicated in neurodegenerative diseases such as *C9ORF72*-associated amyotrophic lateral sclerosis and frontotemporal dementia (C9-ALS/FTD). Here, we identify a neuroprotective pathway of like-Sm protein 12 (*LSM12*) and exchange protein directly activated by cyclic AMP 1 (*EPAC1*) that sustains the nucleocytoplasmic RAN gradient and thereby suppresses NCT dysfunction by the *C9ORF72*-derived poly(glycine-arginine) protein. LSM12 depletion in human neuroblastoma cells aggravated poly(GR)-induced impairment of NCT and nuclear integrity while promoting the nuclear accumulation of poly(GR) granules. In fact, *LSM12* posttranscriptionally up-regulated *EPAC1* expression, whereas EPAC1 overexpression rescued the RAN gradient and NCT defects in *LSM12*-deleted cells. C9-ALS patient-derived neurons differentiated from induced pluripotent stem cells (C9-ALS iPSNs) displayed low expression of *LSM12* and *EPAC1*. Lentiviral overexpression of LSM12 or EPAC1 indeed restored the RAN gradient, mitigated the pathogenic mislocalization of TDP-43, and suppressed caspase-3 activation for apoptosis in C9-ALS iPSNs. EPAC1 depletion biochemically dissociated RAN-importin β1 from the cytoplasmic nuclear pore complex, thereby dissipating the nucleocytoplasmic RAN gradient essential for NCT. These findings define the *LSM12-EPAC1* pathway as an important suppressor of the NCT-related pathologies in C9-ALS/FTD.

## Introduction

Amyotrophic lateral sclerosis (ALS) is a fatal neurodegenerative disease that manifests as progressive loss of motor neurons, paralysis, and respiratory failure [1]. ALS shares pathological hallmarks with frontotemporal dementia (FTD), a clinically distinct neurodegenerative disorder accompanied by behavioral changes and language difficulties [2,3]. Familial ALS/FTD is most commonly caused by the pathogenic expansion of GGGGCC hexanucleotide repeats in the *C9ORF72* locus [4–6]. Additional genetic factors associated with ALS/FTD include TDP-

sequencing experiments can be downloaded from GEO (accession number GSE160159).

**Funding:** This work was supported by grants from the Suh Kyungbae Foundation (SUHF-17020101 [CL]); from the National Research Foundation funded by the Ministry of Science and Information & Communication Technology (MSIT), Republic of Korea (NRF-2018R1A2B2004641[CL]; NRF-2018R1A5A1024261[KJY, YKK, and CL]); and from the Korea Health Technology R&D Project through the KHIDI funded by the Ministry of Health & Welfare, Republic of Korea (HI16C1747[CL]). The funders had no role in study design, data collection and analysis, decision to publish, or preparation of the manuscript.

**Competing interests:** The authors have declared that no competing interests exist.

**Abbreviations:** ALS, amyotrophic lateral sclerosis; ATXN2, ataxin-2; C9-ALS, C9ORF72-associated amyotrophic lateral sclerosis; DEG, differentially expressed gene; DPR, dipeptide repeat; EIF2α, eukaryotic translation initiation factor 2 subunit α; EPAC1, exchange protein directly activated by cyclic AMP 1; FTD, frontotemporal dementia; GFP, green fluorescent protein; HGPS, Hutchinson–Gilford progeria syndrome; IP, immunoprecipitation; iPSC, induced pluripotent stem cell; ISRIB, integrated stress response inhibitor; LCD, low complexity sequence domain; LSM12, like-Sm protein 12; NCT, nucleocytoplasmic transport; NPC, neural progenitor cell; NTF2, nuclear transport factor 2; PAF1, RNA polymerase II-associated factor 1; polyQ, polyglutamine; RANBP2, Ran-binding protein 2; RANGAP1, RAN GTPase-activating protein 1; RANGEF, RAN guanine nucleotide exchange factor; RIN, RNA integrity number; SG, stress granule; sgRNA, small guide RNA; shRNA, short hairpin RNA; siRNA, small interfering RNA; S-tdT, S-tdTomato; SUPT4H1, suppressor of Ty 4 homolog 1; TRAP, translating ribosome affinity purification; UTR, untranslated region.

43 and FUS, the two RNA-binding proteins that form cytoplasmic aggregates in the affected neurons of ALS/FTD subtypes [7]. Nonetheless, the prevalence of sporadic ALS/FTD and their genetic heterogeneity suggest the multifactorial nature of these genetic disorders [8–10] and the substantial contribution of nongenetic factors to ALS/FTD pathogenesis [11,12].

In *C9ORF72*-associated ALS/FTD, the RNA polymerase II-associated factor 1 (PAF1) complex and transcription elongation factor SUPT4H1 (suppressor of Ty 4 homolog 1) promote bidirectional transcription from *C9ORF72*-associated repeats [13,14]. These RNA molecules containing repetitive sequences form pathogenic secondary structures that sequester RNA-binding proteins in a conformation-specific manner and interfere with their relevant function [15,16]. Accordingly, it has been suggested that the gain of RNA toxicity is responsible for the pathogenesis of *C9ORF72*-associated ALS/FTD [17,18]. Furthermore, *C9ORF72* repeats encoded in both sense and antisense RNAs are translated into 5 different dipeptide repeat (DPR) proteins via repeat-associated non-AUG translation of all possible reading frames [19,20]. While all DPR proteins are detected in *C9ORF72*-related ALS/FTD patients [19,21], it has been demonstrated that a subset of DPR proteins (i.e., poly(GR), poly(PR), and poly(GA) proteins) form intracellular inclusions and impair specific aspects of cellular physiology [2,22,23]. Thus, the cytotoxic effects of these noncanonical translation products are emerging as the key mechanism that contributes to neurodegeneration [24].

The arginine-containing DPR proteins, poly(GR) and poly(PR), associate with RNA-binding proteins that display low complexity sequence domains (LCDs) and form membrane-less intracellular organelles via phase separation [25–29]. Poly(GR) and poly(PR) proteins localize to nucleoli, translocate the nucleolar phosphoprotein B23 (also known as NPM1) into the nucleoplasm, and impair both pre-mRNA splicing and ribosomal RNA processing [25,26,30,31]. DPR-induced nucleolar stress is thus emerging as one of the key pathogenic mechanisms underlying *C9ORF72*-associated ALS/FTD [32]. On the other hand, both DPR proteins promote the formation of stress granules (SGs) [25,28,33,34], cytoplasmic assemblies of ribonucleoproteins that are formed under diverse cellular stresses and are thought to support cellular homeostasis of RNA and RNA-binding proteins [35,36]. Genetic studies in yeast, *Drosophila*, and in vitro cell culture models of *C9ORF72*-associated ALS/FTD have further identified several cellular factors involved in nucleocytoplasmic transport (NCT) as genetic modifiers of DPR cytotoxicity [22,37–41]. A compelling model suggests that the nonfunctional sequestration of key NCT factors into DPR-induced SGs disrupts NCT, thereby contributing to neurodegeneration in *C9ORF72*-associated ALS/FTD [33]. More direct effects of poly(GR) and poly(PR) proteins on karyopherin function via biochemical association have also been shown in in vitro assays [41].

Polyglutamine (polyQ) expansions of ataxin-2 (ATXN2) are associated with a high risk of ALS [42]. Transgenic ATXN2 depletion or loss-of-function mutation in *ATXN2* mitigates neurodegenerative phenotypes in genetic disease models of ALS/FTD [33,39,43,44]. Because ATXN2 protein localizes to SGs and is necessary for SG assembly under oxidative stress conditions [45,46], it has been proposed that polyQ expansions cause gain-of-function effects on ATXN2-dependent assembly of SGs, which promotes neurodegeneration. In contrast, loss of *ATXN2* function may suppress neurodegeneration, likely through impairment of SG formation [43,47,48]. Other ALS/FTD-related genetic factors (e.g., TDP-43, FUS, and HNRNPA1) harbor LCDs that similarly regulate SG assembly via liquid–liquid phase separation [49–52]. It has also been shown that C9ORF72 associates with SGs and contributes to SG formation and its clearance via autophagy [53,54]. Loss of *C9ORF72* function thus hypersensitizes the affected cells to cellular stress and is likely involved in the ALS/FTD pathogenesis given that repeat expansion in the *C9ORF72* locus reduces endogenous C9ORF72 expression [53,54].

We previously identified like-Sm protein 12 (LSM12) as an ATXN2-associating adaptor protein that forms a translational activator complex important for circadian rhythms in *Drosophila* [55]. LSM12 recruits the *Drosophila*-specific translation factor, TWENTY-FOUR [56], to the ATXN2 protein complex, induces translation of the circadian clock gene, *period*, and maintains 24-hour periodicity in circadian locomotor behaviors [55,57,58]. Since the biochemical association of LSM12 with ATXN2 is well conserved between *Drosophila* and humans [55], we asked whether LSM12 would cooperate with ATXN2 to facilitate neurodegeneration. Here, we demonstrate an unexpected role of *LSM12* and its downstream effector, exchange protein directly activated by cyclic AMP 1 (*EPAC1*; also known as *RAPGEF3*) in establishing a robust nucleocytoplasmic gradient of RAN-GTP. Consequently, *LSM12-EPAC1* constitutes an *ATXN2*-independent neuroprotective pathway that sustains NCT and suppresses the cellular pathogenesis of poly(GR)-induced neurodegeneration.

## Results

### LSM12 depletion attenuates SG formation upon arsenite-induced oxidative stress

Given that SG assembly is suppressed by loss of *ATXN2* function [43,45,46], we asked if ATXN2-associated LSM12 plays a similar role in SG formation. To better analyze the kinetics of SG assembly, we induced mild oxidative stress in the SH-SY5Y human neuroblastoma cell line with 50-μM arsenite and then visualized the formation of G3BP1- or PABPC1-positive SGs. Under these conditions, SG assembly underwent a gradual maturation process whereby the average size of SGs per cell became larger through fusion over 2 hours after incubation with arsenite (Fig 1A and 1B). LSM12 depletion by stable transfection of short hairpin RNA (shRNA), however, decreased the relative proportion of SG-positive cells, suggesting that loss of *LSM12* function may increase the threshold for initiating SG assembly. Further quantification revealed that each LSM12-depleted cell initially formed fewer SGs than control cells and displayed a smaller average size of SGs during maturation. The SG phenotypes in LSM12-depleted cells were insensitive to puromycin treatment that destabilizes polysomes and promotes SG formation in control cells (S1A Fig) [59]. To further probe *LSM12* effects on SG disassembly, we induced SG formation with a high dose of arsenite (500 μM) for 1 hour and then traced SG disassembly after the removal of arsenite from cell culture media (S1B and S1C Fig). Under the acute oxidative stress, weaker effects of LSM12 depletion on SG formation persisted during SG disassembly and became undetectable 4 hours after recovery. *LSM12* effects on SGs were relatively specific to arsenite-induced oxidative stress since SG formation under sorbitol-induced osmotic stress [60] was comparable between control and LSM12-depleted cells (S2A Fig). These SG phenotypes in LSM1*2*-depleted cells also correlated with lower levels of eukaryotic translation initiation factor 2 subunit α (EIF2α) phosphorylation upon oxidative stress, but not upon other cellular stresses [61,62] (S2B and S2C Fig).

In fact, LSM12 depletion modestly decreased endogenous levels of ATXN2 protein (Fig 1C). Whereas LSM12 may stabilize ATXN2 through the formation of a protein complex, we found that ATXN2 overexpression partially rescued the average number and size of arsenite-induced SGs in LSM12-depleted cells (Fig 1D and 1E). However, a lower percentage of SG-positive cells in LSM12-depleted cells was not evidently rescued by ATXN2 overexpression, indicating more direct effects of *LSM12* on the threshold for initiating SG assembly. ATXN2 depletion, on the other hand, did not affect endogenous levels of LSM12 protein (Fig 1C) but caused an impairment in SG formation similar to that observed in LSM12-depleted cells (Fig 1A and 1B). Moreover, we observed the nonadditive effects of *LSM12* and *ATXN2* on arsenite-

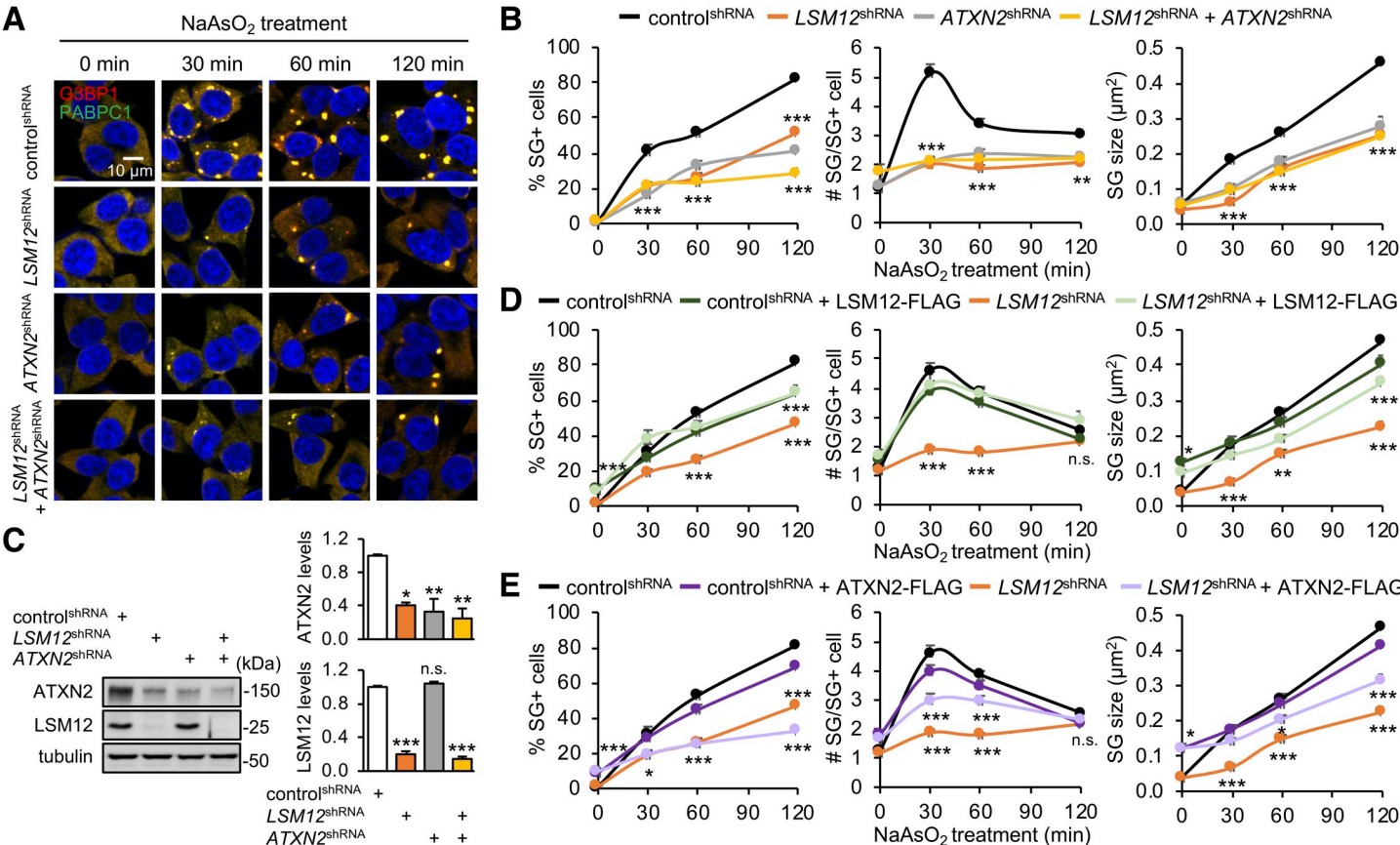

**Fig 1. LSM12 depletion attenuates SG formation upon arsenite-induced oxidative stress.** (A) *LSM12* and *ATXN2* promote arsenite-induced SG assembly, likely via the same genetic pathway. SH-SY5Y cells stably expressing each shRNA were incubated with 50-μM sodium arsenite (NaAsO$_2$) for the indicated time and then co-stained with anti-G3BP1 antibody (red), anti-PABPC1 antibody (green), and Hoechst 33258 (blue) to visualize SGs and the nucleus, respectively. (B) The percentage of SG-positive cells, the number of SGs per SG-positive cell, and the size of SGs were quantified using ImageJ software and averaged (*n* = 15–19 confocal images of random fields of interest obtained from 3 independent experiments; *n* = 329–1,045 cells). Error bars indicate SEM. **$P < 0.01$, ***$P < 0.001$ to control[shRNA] cells at a given time point, as determined by 2-way ANOVA with Tukey post hoc test. (C) Immunoblotting of total cell extracts from individual shRNA cell lines with anti-ATXN2, anti-LSM12, and anti-tubulin (loading control) antibodies. The abundance of each protein was quantified using ImageJ and normalized to that of tubulin. Relative protein levels were then calculated by normalizing to those in control[shRNA] cells. Data represent means ± SEM (*n* = 3). n.s., not significant; *$P < 0.05$, **$P < 0.01$, ***$P < 0.001$, as determined by 2-way ANOVA with Tukey post hoc test. (D, E) Overexpression of LSM12, but not ATXN2, restores arsenite-induced SG assembly in LSM12-depleted cells. Control[shRNA] and *LSM12*[shRNA] cells were transfected with an expression vector for FLAG, FLAG-tagged LSM12, or FLAG-tagged ATXN2. Arsenite-induced SG assembly was quantified 48 hours after transfection. Data represent means ± SEM (*n* = 15–16 confocal images obtained from 3 independent experiments; *n* = 200–820 cells). n.s., not significant; *$P < 0.05$, **$P < 0.01$, ***$P < 0.001$ to control[shRNA] cells expressing FLAG at a given time point, as determined by 2-way ANOVA with Tukey post hoc test. All underlying numerical values are available in S1 Data. ANOVA, analysis of variance; ATXN2, ataxin-2; LSM12, like-Sm protein 12; SEM, standard error of the mean; SG, stress granule; shRNA, short hairpin.

induced SG assembly, suggesting that these factors may act together in the same genetic pathway to regulate SG formation under oxidative stress conditions.

## LSM12 depletion disrupts the RAN gradient and impairs NCT upon oxidative stress

Emerging evidence indicates that SGs sequester cellular factors important for NCT and thereby interfere with NCT under diverse cellular or genetic stresses [33,63]. Consistent with this, it has been shown that inhibition of SG assembly by ATXN2 depletion or treatment with integrated stress response inhibitor (ISRIB) rescues stress-induced disruption of NCT, identifying SG inhibition as a neuroprotective mechanism [33]. Given *LSM12* effects on arsenite-

induced SG assembly, we hypothesized that LSM12 depletion would mitigate the arsenite-induced impairment of NCT. To quantify NCT activity in SH-SY5Y cells, we employed a green fluorescent protein (GFP) reporter harboring both nuclear localization and export signals (shuttle-GFP/S-GFP) that predominantly localizes to the cytoplasm under basal conditions [64]. Any change in the ratio of nuclear to cytoplasmic S-GFP localization upon pharmacological or genetic perturbation would reflect quantitative alterations in NCT function. Consistent with previous findings [33], arsenite-induced oxidative stress significantly increased the nuclear fraction of S-GFP, whereas ATXN2 depletion or ISRIB treatment substantially blocked both SG formation and NCT defects upon oxidative stress (Fig 2A and 2B). Surprisingly, LSM12 depletion facilitated rather than hindered the nuclear translocation of S-GFP upon arsenite treatment (S3A Fig). Moreover, ATXN2 depletion or ISRIB treatment failed to rescue the *LSM12* phenotypes (Fig 2A and 2B). These effects were not specific to the S-GFP reporter since similar NCT defects were observed in LSM12-depleted cells expressing S-tdTomato (S-tdT), an independent NCT reporter that predominantly localizes to the nucleus under basal conditions [37] (S3B Fig). These findings convincingly demonstrate that *LSM12* sustains NCT in a manner independent of *ATXN2* or SG assembly under oxidative stress conditions.

RAN is an evolutionarily conserved, small GTPase that shuttles between the nucleus and cytoplasm in 2 alternating forms: GTP-bound and GDP-bound [65–67]. The opposing activities of RAN GTPase-activating protein 1 (RANGAP1) at the cytoplasmic side of the nuclear pore complex and of chromatin-associating RAN guanine nucleotide exchange factor (RAN-GEF; also known as RCC1), establish a steep nucleocytoplasmic gradient of RAN-GTP [68,69]. The RAN-GTP/GDP state subsequently switches its binding affinity between nuclear transport factors, thereby defining the directional NCT of a given cargo protein via the nuclear pore complex [70]. Disruption of the nucleocytoplasmic RAN gradient has been observed in neurodegenerative diseases, such as *C9ORF72*-ALS, Huntington's disease, and Alzheimer's disease [33,71,72]. We thus asked whether an abnormal RAN gradient would explain LSM12-depletion phenotypes in NCT. In control cells, arsenite-induced oxidative stress increased the relative abundance of cytoplasmic RAN (Fig 2C and 2D). ATXN2 depletion or ISRIB treatment suppressed the arsenite-induced disruption of the RAN gradient, consistent with previous observations [33]. On the other hand, LSM12 depletion itself was sufficient to disrupt the RAN gradient, whereas arsenite treatment had no additive effects on the RAN gradient in LSM12-depleted cells (Fig 2C and 2D). Consistent with our observations in NCT assays, ATXN2 depletion or ISRIB treatment negligibly affected RAN phenotypes caused by LSM12 depletion. It is puzzling that LSM12 depletion disrupts the RAN gradient regardless of arsenite treatment while impairing NCT only under the oxidative stress. We speculate that there may be a compensating mechanism for LSM12 deficiency in non-stressed cells to sustain NCT. In this sense, LSM12 likely acts as a risk rather than a causative factor for NCT-relevant pathogenesis. Taken together, our data suggest that *LSM12* has a role in establishing a basal RAN gradient, and these effects are likely distinct from *LSM12* function in SG assembly.

## LSM12 depletion facilitates the nuclear accumulation of *C9ORF72*-derived poly(GR) protein and exacerbates its pathogenic effects

Given that pathogenic proteins implicated in ALS/FTD disrupt NCT via induction of SG formation [33], we wondered if the loss of *LSM12* function would exacerbate these cellular processes, which are involved in neurodegeneration and possibly normal aging [73]. To test this possibility, we employed *C9ORF72*-derived poly(GR) protein translated from a codon-optimized synthetic cDNA encoding 100 GR repeats [74]. Since poly(GR) protein localizes to

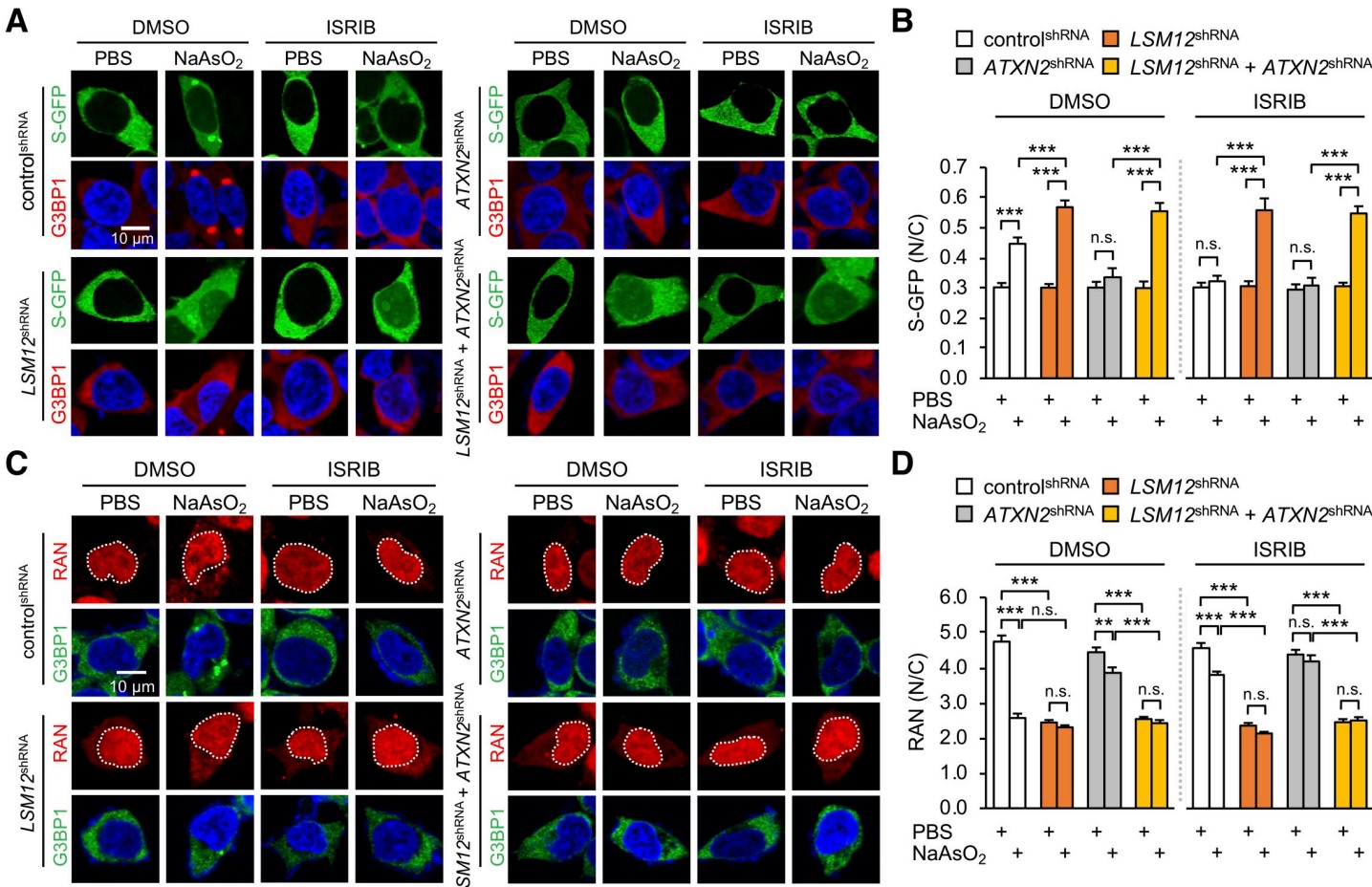

**Fig 2. LSM12 depletion disrupts the RAN gradient and impairs NCT upon oxidative stress.** (A) LSM12 depletion impairs NCT under oxidative stress conditions in a manner that is independent of *ATXN2* or SG assembly. Individual shRNA cell lines were transfected with an expression vector for the S-GFP reporter. SG assembly was then quantified 48 hours after transfection. Transfected cells were co-stained with anti-G3BP1 antibody (red) and Hoechst 33258 (blue). Where indicated, cells were incubated with 50-μM $NaAsO_2$ or PBS (vehicle control) for 2 hours to induce oxidative stress. Phospho-EIF2α–dependent SG assembly was blocked by treating with 2-μM ISRIB for 3 hours before $NaAsO_2$ incubation. DMSO was used as vehicle control for ISRIB. (B) NCT of S-GFP reporter proteins was quantified by calculating the ratio of N/C fluorescence in individual cells. Two-way ANOVA detected significant interaction effects of arsenite and ISRIB treatments on NCT only in control[shRNA] cells ($P = 0.0004$). Data represent means ± SEM ($n = 100$–137 cells from 3 independent experiments). n.s., not significant; ***$P < 0.001$, as determined by Tukey post hoc test. (C) LSM12 depletion disrupts the nucleocytoplasmic RAN gradient in a manner that is independent of *ATXN2* or SG assembly. Cells were incubated with 2-μM ISRIB, 50-μM $NaAsO_2$, or vehicle controls as described above and then co-stained with anti-RAN antibody (red), anti-G3BP1 antibody (green), and Hoechst 33258 (blue). (D) The relative distribution of endogenous RAN proteins was quantified by calculating the ratio of N/C fluorescence. Two-way ANOVA detected significant interaction effects of arsenite and ISRIB treatments on the RAN gradient only in control[shRNA] cells ($P < 0.0001$). Data represent means ± SEM ($n = 95$–104 cells from 3 independent experiments). n.s., not significant; **$P < 0.01$, ***$P < 0.001$, as determined by Tukey post hoc test. All underlying numerical values are available in S1 Data. ANOVA, analysis of variance; ATXN2, ataxin-2; GFP, green fluorescent protein; ISRIB, integrated stress response inhibitor; LSM12, like-Sm protein 12; N/C, nuclear to cytoplasmic; NCT, nucleocytoplasmic transport; SEM, standard error of the mean; SG, stress granule; shRNA, short hairpin.

NPM1-positive nucleoli and causes nucleolar stress [25,26,30], we tested whether *LSM12* contributed to the nucleolar assembly of poly(GR) granules as well as poly(GR)-induced SG. LSM12 depletion significantly increased the relative proportion of cells harboring poly(GR)-induced SGs (Fig 3A and 3B). This contrasted with LSM12-depletion phenotype in that of cells harboring arsenite-induced SGs, further indicating the stress-specific effects of *LSM12* on SG assembly. Nonetheless, the smaller G3BP-positive SGs in LSM12-depleted cells likely indicates a role for *LSM12* in the maturation process of poly(GR)-induced SGs, similar to that observed in arsenite-induced SGs. LSM12 depletion caused more striking effects on the formation of nuclear poly(GR) granules. Control cells gradually accumulated poly(GR) protein in nucleolar

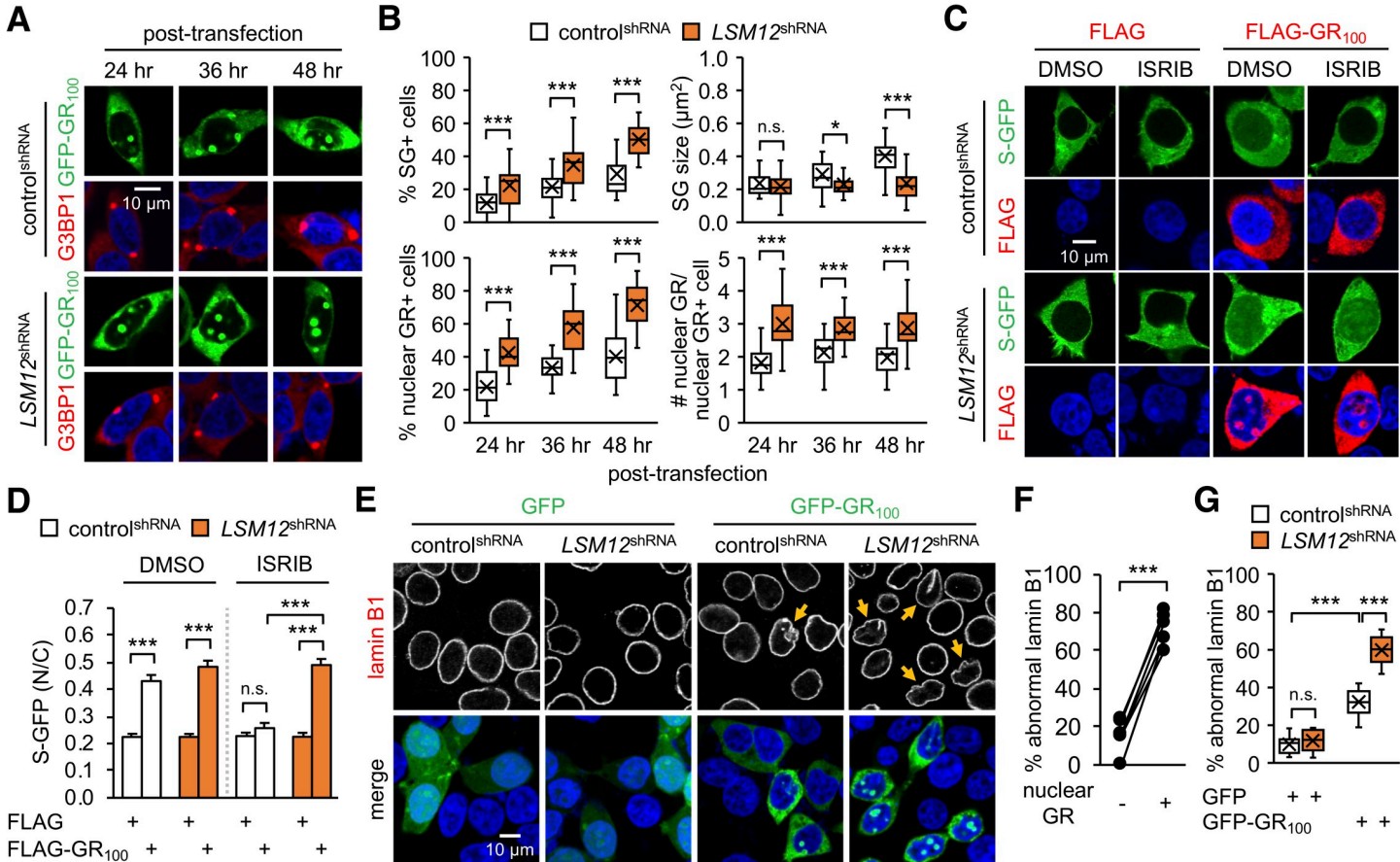

**Fig 3. LSM12 depletion facilitates the nuclear accumulation of *C9ORF72*-derived poly(GR) proteins and exacerbates their pathogenic effects.** (A) LSM12 depletion suppresses the maturation of poly(GR)-induced SGs but promotes the nuclear accumulation of poly(GR) granules. Control[shRNA] and *LSM12*[shRNA] cells were transfected with a GFP-GR$_{100}$ expression vector and then co-stained with anti-G3BP1 antibody (red) and Hoechst 33258 (blue) at the indicated time after transfection. (B) The assemblies of poly(GR)-induced SGs and nuclear poly(GR) granules were quantified as in Fig 1. Data represent means ± SEM (*n* = 23–25 confocal images obtained from 3 independent experiments; *n* = 424–513 GFP-GR$_{100}$–positive cells). n.s., not significant; *$P < 0.05$, ***$P < 0.001$, as determined by Student *t* test. (C) ISRIB treatment suppresses the poly(GR)-induced disruption of NCT in control cells, but not in LSM12-depleted cells. Control[shRNA] and *LSM12*[shRNA] cells were co-transfected with expression vectors for S-GFP and FLAG-GR$_{100}$. Where indicated, cells were incubated with 2-μM ISRIB or DMSO (vehicle control) for 5 hours and then co-stained with anti-FLAG antibody (red) and Hoechst 33258 (blue) 48 hours after transfection. (D) NCT of S-GFP reporter proteins was quantified as in Fig 2B. Data represent means ± SEM (*n* = 142–166 cells from 3 independent experiments). n.s., not significant; ***$P < 0.001$, as determined by 2-way ANOVA with Tukey post hoc test. (E) LSM12 depletion exacerbates poly(GR)-induced disruption of the nuclear lamina. Control [shRNA] and *LSM12*[shRNA] cells were transfected with an expression vector for GFP or GFP-GR$_{100}$ and then co-stained with anti-lamin B1 antibody (red) and Hoechst 33258 (blue) to visualize nuclear envelope morphology 48 hours after transfection. Yellow arrows indicate GFP-GR$_{100}$–positive cells with severe disruption of the nuclear lamina. (F) Control cells expressing GFP-GR$_{100}$ were scored for nuclear poly(GR) granules and abnormal morphology of the nuclear lamina. The relative percentages of cells with severe nuclear laminar disruption were averaged from confocal images of 6 random fields of interest per condition (*n* = 55–69 GFP-GR$_{100}$–positive cells from 3 independent experiments). Error bars indicate SEM. ***$P < 0.001$, as determined by Student *t* test. (G) The relative percentages of control[shRNA] and *LSM12*[shRNA] cells with severe nuclear lamina disruption were quantified as described above. Data represent means ± SEM (*n* = 10 confocal images obtained from 3 independent experiments; *n* = 183–297 GFP–or GFP-GR$_{100}$–positive cells). n.s., not significant; ***$P < 0.001$, as determined by 2-way ANOVA with Tukey post hoc test. All underlying numerical values are available in S1 Data. ANOVA, analysis of variance; GFP, green fluorescent protein; LSM12, like-Sm protein 12; N/C, nuclear to cytoplasmic; NCT, nucleocytoplasmic transport; SEM, standard error of the mean; SG, stress granule; shRNA, short hairpin.

granules after the transient transfection of cells with the poly(GR) expression vector (Fig 3A and 3B). LSM12 depletion remarkably facilitated this process and increased the number of nuclear poly(GR) granules per cell.

We next examined if these LSM12-depletion phenotypes led to any alterations in poly(GR) toxicity. Overexpression of poly(GR) protein in control cells disrupted NCT in an ISRIB-sensitive manner (Fig 3C and 3D), supporting that poly(GR) effects on NCT require SG formation [33]. By contrast, ISRIB treatment failed to rescue poly(GR)-induced impairment of NCT in

LSM12-depleted cells, an effect similar to that of *LSM12* on NCT under oxidative stress conditions. We further found that poly(GR) overexpression disrupted the integrity of the nuclear envelope (Fig 3E). Abnormalities of the nuclear lamina induced by poly(GR) overexpression were more severe in LSM12-depleted cells than control cells, indicating a strong correlation between the presence of nuclear poly(GR) granules and abnormal morphology of the nuclear envelop (Fig 3F and 3G). We observed comparable phenotypes in a cell line harboring a CRISPR/Cas9-mediated deletion in the *LSM12* genetic locus (S4 Fig). Taken together, these results suggest that *LSM12* delays the nucleolar deposition of poly(GR) protein, thereby suppressing its cytotoxic effects.

## An *LSM12*^V135I^ mutant allele exhibits dominant-negative effects on the RAN gradient

Large-scale genome-wide association studies have revealed an abundance of low-frequency genetic variants associated with ALS patients [8–10]. This is also the case for ALS-associated *LSM12* loci, where several rare point mutations in the *LSM12* coding sequence were detected (http://databrowser.projectmine.com/). Given our observation that *LSM12* suppresses the nuclear assembly of ALS-relevant poly(GR) protein and its pathogenic effects, we hypothesized that some of these *LSM12* variants might display loss-of-function phenotypes, explaining their presence in ALS patients. We, therefore, overexpressed either wild-type or point mutants of LSM12 in control or *LSM12*-deleted cells and assessed their NCT-related functional activities.

We first confirmed that disruption of NCT upon arsenite-induced oxidative stress was more severe in *LSM12*-deleted cells than control cells (Fig 4A and 4B), consistent with LSM12-depletion phenotypes (Fig 2A and 2B). Overexpression of wild-type LSM12 rescued *LSM12*-deletion phenotypes in NCT, whereas LSM12^V135I^, one of the *LSM12* variants observed in ALS patients, failed to do so. In fact, LSM12^V135I^ overexpression was sufficient to impair NCT in both control and *LSM12*-deleted cells regardless of arsenite treatment (Fig 4A and 4B). Since *LSM12* deletion did not affect NCT in control cells not treated with arsenite, LSM12^V135I^ might have neomorphic effects on NCT possibly via genetic induction of oxidative stress. Overexpression of LSM12^V135I^, but not wild-type LSM12, also promoted the assembly of nuclear poly(GR) granules in control cells and increased the cell population that displayed poly(GR)-induced SGs (Fig 4C and 4D).

Finally, we found that these NCT phenotypes were consistent with *LSM12* effects on the RAN gradient. Cytoplasmic mislocalization of RAN in *LSM12*-deleted cells was restored by overexpression of wild-type LSM12 (Fig 4E and 4F). In contrast, LSM12^V135I^ overexpression potently disrupted the RAN gradient in control cells, but it did not exaggerate the RAN phenotype in *LSM12*-deleted cells. Considering that overexpression of neither wild-type LSM12 nor LSM12^V135I^ altered endogenous levels of LSM12 proteins in control cells (S5 Fig), these results indicate the dominant-negative effects of *LSM12*^V135I^ mutant on establishing a RAN gradient. The *LSM12*^V135I^ variant is detectable in other general databases for single nucleotide polymorphism in humans (e.g., https://www.ncbi.nlm.nih.gov/snp/rs1162569843). However, its low allele frequency did not allow us to determine whether or not this mutation is exclusively associated with ALS. Nonetheless, our genetic analyses implicate *LSM12* as a risk factor for NCT-relevant pathogenesis in ALS/FTD.

## *LSM12* posttranscriptionally up-regulates *EPAC1* expression to sustain the RAN gradient for NCT and suppress poly(GR) toxicity

To elucidate how *LSM12* contributes to the establishment of a nucleocytoplasmic RAN gradient, we performed gene expression analyses using RNA sequencing. Given the role of LSM12

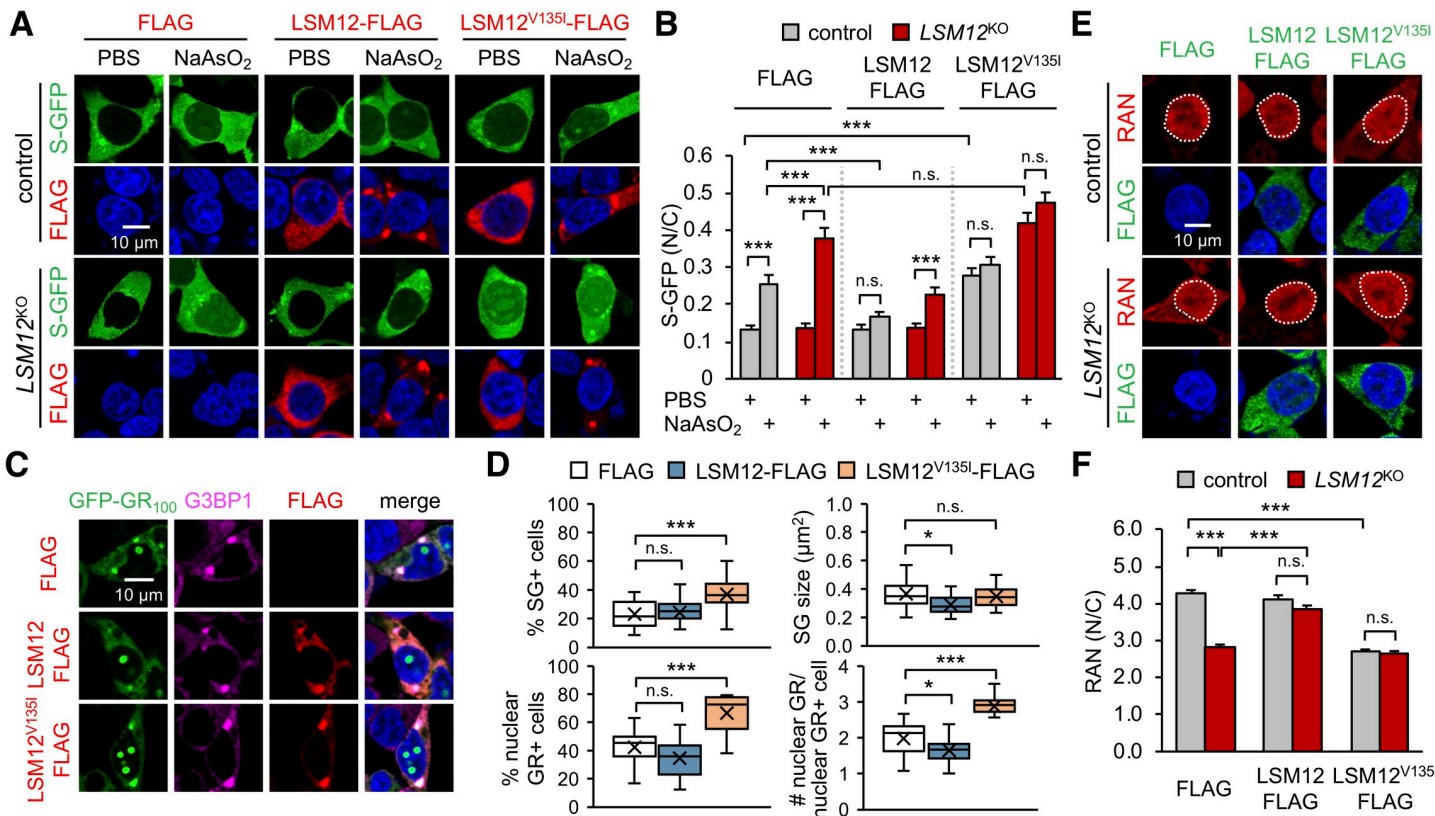

**Fig 4. An *LSM12*^V135I mutant allele exhibits dominant-negative effects on the RAN gradient.** (A) Overexpression of LSM12^V135I mutant protein impairs NCT. Control and *LSM12*^KO cells were co-transfected with expression vectors for S-GFP and FLAG-tagged LSM12 (wild-type or LSM12^V135I mutant). Transfected cells were treated with 50-μM NaAsO$_2$ or PBS (vehicle control) for 2 hours and then co-stained with anti-FLAG antibody (red) and Hoechst 33258 (blue) 48 hours after transfection. (B) NCT of S-GFP reporter proteins was quantified as in Fig 2B. Two-way ANOVA detected significant interaction effects of arsenite treatment and *LSM12* deletion on NCT only in FLAG-expressing cells ($P = 0.0026$). Data represent means ± SEM ($n = 119–138$ cells from 3 independent experiments). n.s., not significant; ***$P < 0.001$, as determined by Tukey post hoc test. (C) Overexpression of LSM12^V135I mutant protein promotes the nuclear accumulation of poly(GR) granules. SH-SY5Y cells were co-transfected with expression vectors for GFP-GR$_{100}$ and FLAG-tagged LSM12 (wild-type or LSM12^V135I mutant) and then co-stained with anti-FLAG antibody (red), anti-G3BP1 antibody (magenta), and Hoechst 33258 (blue) 48 hours after transfection. (D) The assemblies of poly(GR)-induced SGs and nuclear poly(GR) granules were quantified as in Fig 1. Data represent means ± SEM ($n = 19$ confocal images obtained from 3 independent experiments; $n = 279–327$ GFP-GR$_{100}$–positive cells). n.s., not significant; *$P < 0.05$, ***$P < 0.001$, as determined by 1-way ANOVA with Dunnett post hoc test. (E) Overexpression of LSM12^V135I mutant protein disrupts the nucleocytoplasmic RAN gradient. Control and *LSM12*^KO cells were transfected with an expression vector for FLAG-tagged LSM12 (wild-type or LSM12^V135I mutant) and then co-stained with anti-RAN antibody (red), anti-FLAG antibody (green), and Hoechst 33258 (blue) 48 hours after transfection. (F) The nucleocytoplasmic RAN gradient was quantified as in Fig 2D. Data represent means ± SEM ($n = 119–120$ cells from 3 independent experiments). n.s., not significant; ***$P < 0.001$, as determined by 2-way ANOVA with Tukey post hoc test. All underlying numerical values are available in S1 Data. ANOVA, analysis of variance; GFP, green fluorescent protein; LSM12, like-Sm protein 12; SEM, standard error of the mean; N/C, nuclear to cytoplasmic; NCT, nucleocytoplasmic transport.

in translation [55], we compared *LSM12*-dependent changes in total mRNAs with changes in translating ribosome-associated mRNAs (Fig 5A, S6 Fig). Among genes that were differentially expressed between control and LSM12-depleted cells, *EPAC1* displayed low expression in either RNA analyses of LSM12-depleted cells. Further independent analyses confirmed that both *EPAC1* transcript and EPAC1 protein were expressed at low levels in either LSM12-depleted cells or *LSM12*-deleted cells compared with those in control cells (Fig 5B and 5C, S7A Fig). To investigate how *LSM12* up-regulates *EPAC1* expression, we generated *EPAC1* reporter transgenes of which expression was transcriptionally driven by *EPAC1* promoter or posttranscriptionally controlled by *EPAC1* untranslated regions (UTRs) (S7B Fig). While the loss of *LSM12* function did not affect the *EPAC1* promoter activity, we found that *EPAC1* 5′ UTR was necessary and sufficient for *LSM12*-dependent expression of the posttranscriptional *EPAC1*

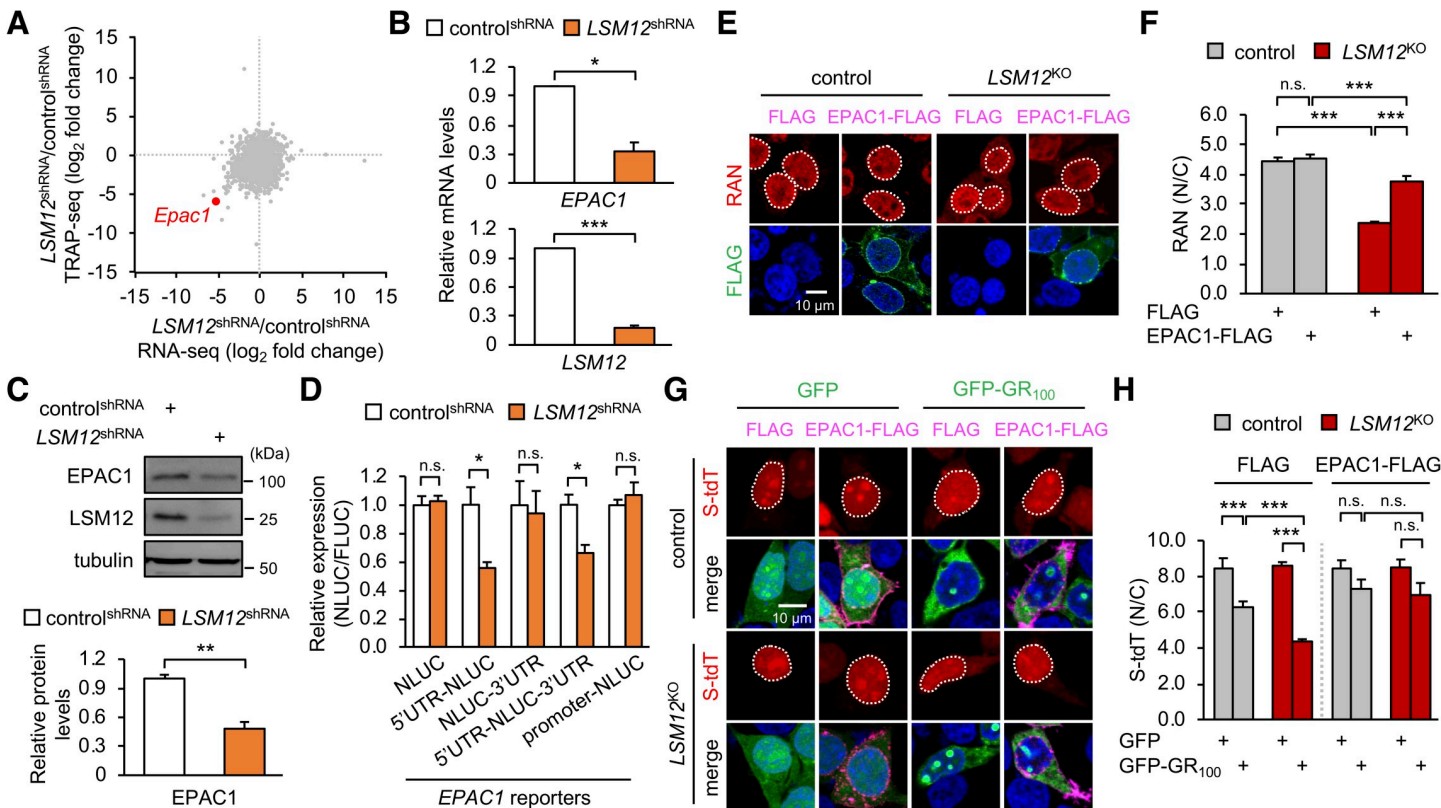

**Fig 5. *LSM12* posttranscriptionally up-regulates *EPAC1* expression to sustain the RAN gradient for NCT.** (A) Fold changes in total transcript levels (x-axis) versus translating ribosome-associated transcript levels (y-axis) in LSM12-depleted cells were assessed by RNA sequencing ($n = 2$ biological replicates; $n = 10,856$ transcripts) and depicted as a scatter plot. (B) LSM12-depleted cells express low levels of *EPAC1* transcript. Total RNA was prepared from control[shRNA] and *LSM12*[shRNA] cells. The abundance of each transcript was quantified by real-time RT-PCR and normalized to that of *GAPDH*. Relative mRNA levels in *LSM12*[shRNA] cells were then calculated by normalizing to those in control[shRNA] cells. Data represent means ± SEM ($n = 3$). *$P < 0.05$, ***$P < 0.001$, as determined by Student $t$ test. (C) LSM12-depleted cells express low levels of EPAC1 protein. The abundance of each protein was quantified as in Fig 1C. Data represent means ± SEM ($n = 3$). **$P < 0.01$, as determined by Student $t$ test. (D) LSM12 depletion posttranscriptionally decreases *EPAC1* expression via the 5′ UTR. *EPAC1* reporter plasmids encoding NLUC were generated as depicted in S7B Fig. Control[shRNA] and *LSM12*[shRNA] cells were co-transfected with each *EPAC1* reporter and FLUC expression vector (normalizing control). Luciferase reporter assays were performed 48 hours after transfection. NLUC activity was first normalized to FLUC activity per condition. Relative expression of each *EPAC1* reporter in *LSM12*[shRNA] cells was then calculated by normalizing to the NLUC/FLUC value in control[shRNA] cells. Data represent means ± SEM ($n = 4$). n.s., not significant; *$P < 0.05$, as determined by Student $t$ test. (E) EPAC1 overexpression restores the nucleocytoplasmic RAN gradient in *LSM12*-deleted cells. Control and *LSM12*[KO] cells were transfected with an expression vector for FLAG or FLAG-tagged EPAC1 and then co-stained with anti-RAN antibody (red), anti-FLAG antibody (green), and Hoechst 33258 (blue) 48 hours after transfection. (F) The nucleocytoplasmic RAN gradient was quantified as in Fig 2D. Data represent means ± SEM ($n = 103$–$107$ cells from 3 independent experiments). n.s., not significant; ***$P < 0.001$, as determined by 2-way ANOVA with Tukey post hoc test. (G) EPAC1 overexpression suppresses *LSM12*-deletion effects on the poly(GR)-induced disruption of NCT. Control and *LSM12*[KO] cells were co-transfected with different combinations of expression vectors for S-tdT, GFP-GR$_{100}$, and FLAG-tagged EPAC1 and then co-stained with anti-FLAG antibody (magenta) and Hoechst 33258 (blue) 48 hours after transfection. (H) NCT of S-tdT reporter proteins was quantified as in Fig 2B. Two-way ANOVA detected significant interaction effects of GFP-GR$_{100}$ and *LSM12* deletion on NCT only in FLAG-expressing cells ($P = 0.0018$). Data represent means ± SEM ($n = 123$–$153$ cells from 3 independent experiments). n.s., not significant; ***$P < 0.001$, as determined by Tukey post hoc test. All underlying numerical values are available in S1 Data. ANOVA, analysis of variance; EPAC1, exchange protein directly activated by cyclic AMP 1; FLUC, firefly luciferase; GFP, green fluorescent protein; LSM12, like-Sm protein 12; N/C, nuclear to cytoplasmic; NCT, nucleocytoplasmic transport; NLUC, Nano-luciferase; RNA-seq, RNA sequencing; RT-PCR, reverse transcription PCR; S-tdT, S-tdTomato; SEM, standard error of the mean; shRNA, short hairpin; TRAP-seq, translating ribosome affinity purification sequencing.

reporters (Fig 5D, S7C Fig). These results thus validate that *LSM12* acts as a posttranscriptional activator of *EPAC1*.

EPAC1 has been shown to act as a cAMP sensor for the activation of RAP signaling on the plasma membrane [75]. On the other hand, EPAC1 also localizes to the nuclear envelope and associates with nucleoporin complexes containing RAN, RANGAP1, Ran-binding protein 2 (RANBP2; also known as NUP358), and importin β1 [75–77]. We thus hypothesized that the action of EPAC1 in nucleoporin assembly or function might contribute to *LSM12* effects on

the RAN gradient and NCT. To test this possibility, we first examined if EPAC1 overexpression restored the RAN gradient in *LSM12*-deleted cells. Indeed, EPAC1 overexpression substantially suppressed the aberrant localization of RAN in the cytoplasm of *LSM12*-deleted cells, while negligibly affecting the RAN gradient in control cells (Fig 5E and 5F). Consistent with this rescue effect, EPAC1 overexpression suppressed loss-of-function effects of *LSM12* on the disruption of NCT induced by poly(GR) overexpression (Fig 5G and 5H). These data indicate that EPAC1 is limiting for sustaining the RAN gradient and NCT in *LSM12*-deleted cells.

Next, we asked whether the loss of *EPAC1* function would be sufficient to induce cellular phenotypes observed with the loss of *LSM12* function. To this end, we depleted endogenous EPAC1 protein by transiently transfecting SH-SY5Y cells with small interfering RNA (siRNA) targeting the *EPAC1* transcript (S8A Fig). EPAC1 depletion indeed decreased the ratio of nuclear to cytoplasmic RAN distribution regardless of poly(GR) overexpression (Fig 6A and 6B), mimicking the RAN phenotypes in LSM12-depleted or *LSM12*-deleted cells under basal conditions. Moreover, we observed nonadditive disruption of the RAN gradient by EPAC1 depletion and *LSM12* deletion (S8B Fig), further confirming that EPAC1 and LSM12 act in the same pathway for sustaining a RAN gradient. ISRIB treatment failed to suppress the effects of EPAC1 depletion on the RAN gradient (Fig 6A and 6B) and poly(GR)-induced disruption of NCT (S8C and S8D Fig), consistent with an SG-independent role of *LSM12* on the RAN gradient and NCT. Not surprisingly, EPAC1 depletion enhanced nuclear accumulation of poly(GR) protein (Fig 6C and 6D) and increased the poly(GR)-expressing cell population harboring SGs (S8E Fig). Consequently, nuclear membrane integrity was severely impaired by poly(GR) overexpression in EPAC1-depleted cells in an ISRIB-insensitive manner (Fig 6E and 6F). Finally, either heterozygosity of *Epac* deletion mutation [78] or RNA interference-mediated depletion of EPAC exacerbated degeneration of *Drosophila* photoreceptor neurons overexpressing poly (GR) protein (S9 Fig), revealing a conserved role of *EPAC1* homologs in the poly(GR)-induced pathogenesis.

## Overexpression of LSM12 or EPAC1 rescues NCT-relevant pathologies in ALS patient-derived neurons

Our genetic analyses of the loss-of-function phenotypes indicate that deficits in the *LSM12-EPAC1* pathway disrupt the RAN gradient, making the affected cells more susceptible to genetic or environmental perturbations in NCT-related physiology. Nonetheless, we reasoned that a neuroprotective function of this pathway would be better demonstrated if its up-regulation could rescue ALS/FTD-relevant pathologies in the RAN gradient and NCT. We first tested this possibility in SH-SY5Y cells transiently expressing poly(GR) protein. Indeed, overexpression of LSM12 or EPAC1 significantly suppressed poly(GR)-induced disruption of the RAN gradient (Fig 6G and 6H) and nuclear integrity (Fig 6I and 6J).

To validate the disease relevance of these findings, we employed induced pluripotent stem cell (iPSC)-derived neurons from 3 *C9ORF72*-associated ALS patients (C9-ALS iPSNs). A previous study showed that all these C9-ALS iPSC lines (CS28, CS29, and CS52) had mutant *C9ORF72* alleles with approximately 800 hexanucleotide repeats [79]. In addition, an isogenic control iPSC line (CS29-ISO) was established by the CRISPR/Cas9-mediated deletion of the hexanucleotide repeat expansion in one of these C9-ALS iPSC lines (C9-ALS CS29) [80]. This isogenic pair (CS29-ISO and C9-ALS CS29) accordingly served as a good resource to compare any effects of the *C9ORF72* hexanucleotide repeats and our lentiviral transgenes in the same genetic background. Immunofluorescence analyses confirmed comparable neuronal differentiation from CS29-ISO and C9-ALS CS29 iPSC lines (S10A Fig). Our experimental conditions led to mixed neuronal cultures, including approximately 95% MAP2-positive (a neuronal

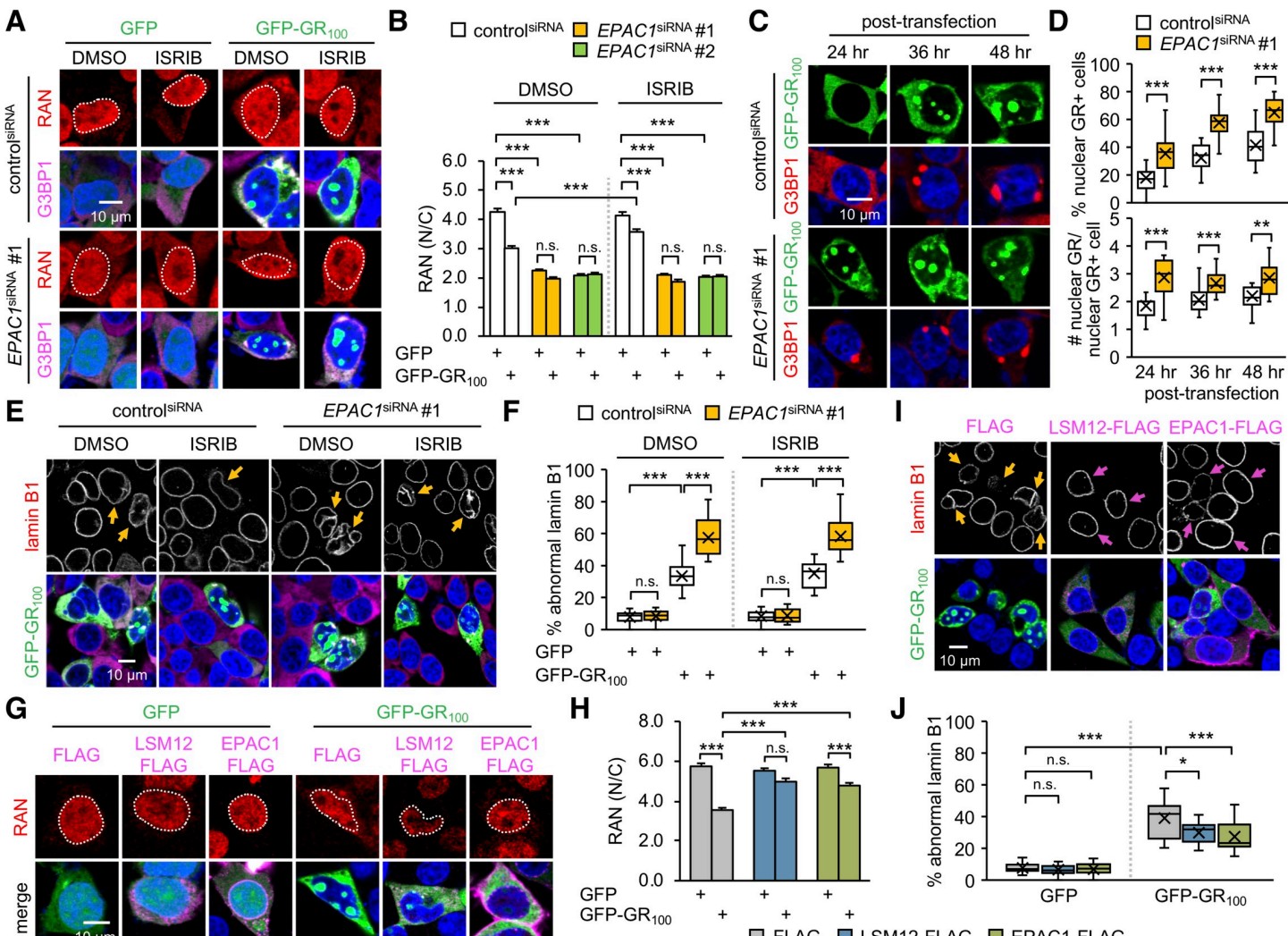

**Fig 6. EPAC1 suppresses poly(GR) toxicity relevant to NCT and nuclear integrity.** (A) SH-SY5Y cells were transfected with control[siRNA] or *EPAC1*[siRNA] 24 hours before transfecting with an expression vector for GFP or GFP-GR$_{100}$. Transfected cells were co-stained with anti-RAN antibody (red), anti-G3BP1 antibody (magenta), and Hoechst 33258 (blue) 72 hours after siRNA transfection. Where indicated, cells were treated with 2-μM ISRIB or DMSO (vehicle control) for 5 hours before antibody staining. (B) The nucleocytoplasmic RAN gradient was quantified as in Fig 2D. Two-way ANOVA detected significant interaction effects of GFP-GR$_{100}$ and ISRIB treatment on the RAN gradient in control[siRNA] cells ($P = 0.0008$), but not in *EPAC1*[siRNA] cells ($P = 0.4674$ for *EPAC1*[siRNA] #1; $P = 0.7870$ for *EPAC1*[siRNA] #2); significant interaction effects of GFP-GR$_{100}$ and EPAC1 depletion on the RAN gradient regardless of ISRIB treatment ($P < 0.0001$ for both *EPAC1*[siRNA] in DMSO; $P = 0.0327$ for *EPAC1*[siRNA] #1 in ISRIB; $P = 0.0013$ for *EPAC1*[siRNA] #2 in ISRIB). Data represent means ± SEM ($n = 124$–145 GFP– or GFP-GR$_{100}$–positive cells from 3 independent experiments). n.s., not significant; ***$P < 0.001$, as determined by Tukey post hoc test. (C) EPAC1 depletion facilitates the nuclear accumulation of poly(GR) protein. SH-SY5Y cells were co-transfected with each siRNA and a GFP-GR$_{100}$ expression vector as above. Transfected cells were co-stained with anti-G3BP1 antibody (red) and Hoechst 33258 (blue) at the indicated time points after plasmid DNA transfection. (D) The assembly of nuclear poly(GR) granules was quantified as in Fig 1. Data represent means ± SEM ($n = 18$–19 confocal images obtained from 3 independent experiments; $n = 366$–413 GFP-GR$_{100}$–positive cells). **$P < 0.01$, ***$P < 0.001$, as determined by Student $t$ test. (E) EPAC1 depletion exacerbates the poly(GR)-induced disruption of the nuclear lamina. SH-SY5Y cells were co-transfected with each siRNA and a GFP-GR$_{100}$ expression vector, treated with 2-μM ISRIB or DMSO (vehicle control) and then co-stained with anti-lamin B1 antibody (red), anti-G3BP1 antibody (magenta), and Hoechst 33258 (blue) as described above. Yellow arrows indicate GFP-GR$_{100}$–positive cells with severe nuclear lamina disruption. (F) The abnormal nuclear laminar morphology was quantified as in Fig 3E. Two-way ANOVA detected significant interaction effects of GFP-GR$_{100}$ and EPAC1 depletion on the nuclear integrity regardless of ISRIB treatment ($P < 0.0001$ for both DMSO and ISRIB). Data represent means ± SEM ($n = 15$ confocal images obtained from 3 independent experiments; $n = 313$–545 GFP– or GFP-GR$_{100}$–positive cells). n.s., not significant; ***$P < 0.001$, as determined by Tukey post hoc test. (G, H) Overexpression of LSM12 or EPAC1 suppresses the poly(GR)-induced disruption of the RAN gradient. SH-SY5Y cells were co-transfected with different combinations of expression vectors for FLAG-tagged LSM12, FLAG-tagged EPAC1, and GFP-GR$_{100}$. Transfected cells were co-stained with anti-RAN antibody (red), anti-FLAG antibody (magenta), and Hoechst 33258 (blue). The nucleocytoplasmic RAN gradient was quantified as in Fig 2D. Two-way ANOVA detected significant interaction effects of GFP-GR$_{100}$ with LSM12 or EPAC1 overexpression on the RAN gradient ($P < 0.0001$ for both). Data represent means ± SEM ($n = 101$–109 GFP– or GFP-GR$_{100}$–positive cells from 3 independent experiments). n.s., not significant; ***$P < 0.001$, as determined by Tukey post hoc test. (I, J) Overexpression of LSM12 or EPAC1 suppresses the poly(GR)-induced disruption of the nuclear lamina. Yellow arrows indicate GFP-GR$_{100}$–positive cells with severe nuclear lamina disruption. Magenta arrows indicate cells overexpressing LSM12-FLAG or EPAC1-FLAG protein. The abnormal morphology of the nuclear lamina was quantified as in Fig 3E. Data represent means ± SEM

($n$ = 11–19 confocal images obtained from 3 independent experiments; $n$ = 235–508 GFP–or GFP-GR$_{100}$–positive cells). n.s., not significant; *$P < 0.05$, ***$P < 0.001$, as determined by 2-way ANOVA with Tukey post hoc test. All underlying numerical values are available in S1 Data. ANOVA, analysis of variance; EPAC1, exchange protein directly activated by cyclic AMP 1; GFP, green fluorescent protein; ISRIB, integrated stress response inhibitor; LSM12, like-Sm protein 12; N/C, nuclear to cytoplasmic; NCT, nucleocytoplasmic transport; SEM, standard error of the mean; siRNA, small interfering RNA.

marker), 70% ChAT-positive (cholinergic neurons), and 25% to 30% HB9-positive cells (motor neurons), as reported previously [81]. Moreover, iPSNs from C9-ALS CS29, but not their isogenic control iPSNs, displayed poly(GR) aggregates at readily detectable levels (S10B Fig).

We further found that the abundance of LSM12 and EPAC1 proteins was significantly decreased in C9-ALS iPSNs from all 3 patients, compared to control iPSNs (Fig 7A). Quantitative transcript analyses revealed lower levels of *LSM12* and *EPAC1* mRNAs in C9-ALS iPSNs (Fig 7B). While the underlying mechanism remains to be determined, we hypothesized that this down-regulation of the *LSM12-EPAC1* pathway might be limiting for NCT-relevant pathologies in C9-ALS iPSNs [33,37,39,81]. A significant reduction (approximately 35%) in the nuclear to cytoplasmic ratio of RAN protein in C9-ALS iPSNs was partially, but significantly, rescued by lentiviral overexpression of LSM12 or EPAC1 (Fig 7C, 7D, S11A Fig). We confirmed that their overexpression negligibly affected neuronal differentiation efficiency per se (S11B and S11C Fig). TDP-43 pathogenesis is the most prominent feature in ALS/FTD, and it has been shown that the loss of nuclear function, as well as the gain of cytoplasmic function, contribute to the underlying neurodegenerative processes [63,82–87]. Given that the nuclear import of TDP-43 is RAN dependent [85,86], we further assessed the neuroprotective effects of the *LSM12-EPAC1* pathway on the cytoplasmic mislocalization of TDP-43 in C9-ALS iPSNs [37]. Consistent with the RAN gradient rescue, overexpression of LSM12 or EPAC1 suppressed TDP-43 mislocalization to the cytoplasm of C9-ALS iPSNs (Fig 7E and 7F). On either the RAN gradient or TDP-43 mislocalization, LSM12$^{V135I}$ overexpression exhibited dominant-negative effects in control iPSNs (Fig 7 C–F).

Finally, we asked whether the transgenic enhancement of the *LSM12-EPAC1* pathway could alleviate neurodegenerative phenotypes in C9-ALS iPSNs. Consistent with previous observations [88,89], the percentages of neurons expressing cleaved caspase-3 were substantially elevated in C9-ALS iPSNs from all 3 patients (Fig 7G and 7H, S12 Fig), likely indicating their pathogenic activation of the proapoptotic pathway. We further found that overexpression of LSM12 or EPAC1 partially, but significantly, suppressed the caspase-3 activation for apoptosis in all 3 C9-ALS iPSN lines (Fig 7G and 7H, S12 Fig). Taken together, these lines of evidence convincingly support our conclusion that LSM12 and EPAC1 constitute a neuroprotective pathway for sustaining the RAN gradient and NCT in the pathophysiology of *C9ORF72*-associated ALS/FTD.

## The *LSM12-EPAC1* pathway promotes RAN-importin β1 loading onto RANBP2-RANGAP1 at the nuclear pore complex

Previous studies have suggested that RANBP2-RANGAP1 recruits the RAN-importin β1 complex to the cytoplasmic side of the nuclear pore complex, facilitating its recycling across the nuclear membrane [90,91]. RAN-GTP actually promotes the association of importin β1 with RANBP2, whereas RAN-GDP loses its affinity for both proteins [90,92,93]. Moreover, EPAC1 associates preferentially with RAN-GTP, and EPAC1 overexpression stabilizes the association between RAN and RANBP2 [75]. We thus hypothesized that the *LSM12-EPAC1* pathway might control the dynamic assembly of the RAN-associating protein complex, thereby contributing to the establishment of the nucleocytoplasmic RAN gradient. To examine this possibility,

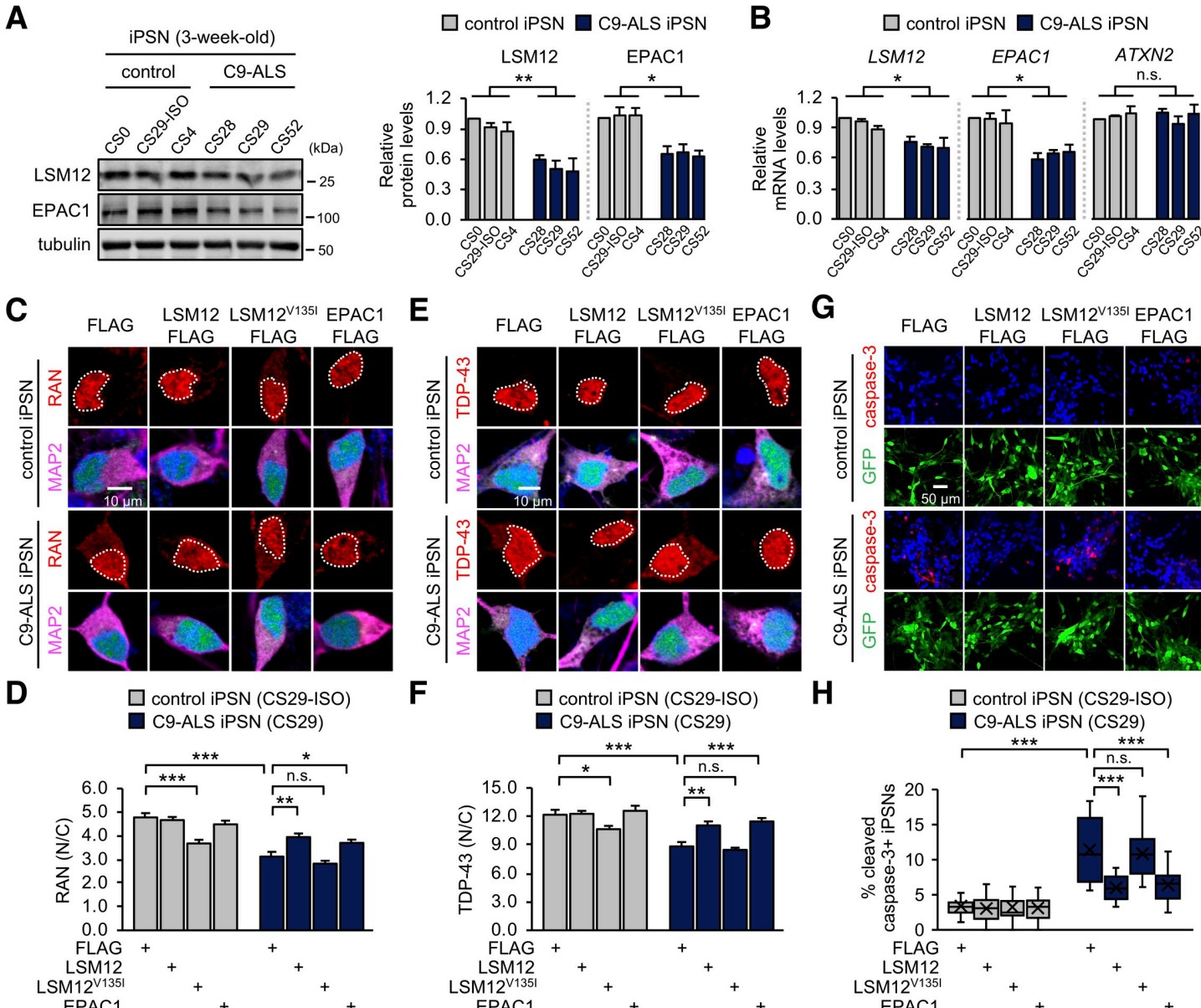

**Fig 7. Overexpression of LSM12 or EPAC1 rescues NCT-relevant pathologies in C9-ALS patient-derived neurons.** (A) C9-ALS iPSNs express low levels of LSM12 and EPAC1 proteins. C9-ALS iPSNs (CS28, CS29, and CS52) and control iPSNs (CS0, CS29-ISO, CS4) were harvested 21 days after neuronal differentiation from NPCs. Total cell extracts from 3-week-old iPSNs were resolved by SDS-PAGE and immunoblotted with anti-LSM12, anti-EPAC1, and anti-tubulin (loading control) antibodies. The abundance of each protein was quantified as in Fig 1C. Error bars indicate SEM ($n$ = 3 independent differentiation experiments). $^*P < 0.05$, $^{**}P < 0.01$, as determined by 1-way ANOVA with Dunnett post hoc test. (B) C9-ALS iPSNs express low levels of *LSM12* and *EPAC1* transcripts. Total RNA was prepared from 3-week-old iPSNs, and the abundance of each transcript was quantified as in Fig 5B. Data represent means ± SEM ($n$ = 3 independent differentiation experiments). n.s., not significant; $^*P < 0.05$, as determined by 1-way ANOVA with Dunnett post hoc test. (C, D) Overexpression of LSM12 or EPAC1 rescues the RAN gradient in C9-ALS iPSNs. NPCs from C9-ALS iPSCs (CS29) and their isogenic control cells (CS29-ISO) were transduced with individual recombinant lentiviruses that express the indicated FLAG-tagged proteins along with a GFP reporter. iPSNs were fixed 21 days after neuronal differentiation from NPCs and co-stained with anti-RAN antibody (red), anti-MAP2 antibody (magenta), and Hoechst 33258 (blue). The nucleocytoplasmic RAN gradient was quantified as in Fig 2D. Two-way ANOVA detected significant interaction effects of C9-ALS and lentiviral overexpression on the RAN gradient ($P$ = 0.0051 for LSM12; $P$ = 0.0066 for LSM12$^{V135I}$; $P$ = 0.0041 for EPAC1). Data represent means ± SEM ($n$ = 100–105 GFP–positive cells from 4 independent differentiation experiments). n.s., not significant; $^*P < 0.05$, $^{**}P < 0.01$, $^{***}P < 0.001$, as determined by Tukey post hoc test. (E, F) Overexpression of LSM12 or EPAC1 suppresses the pathogenic mislocalization of TDP-43 in the cytoplasm of C9-ALS iPSNs. Three-week-old iPSNs (CS29-ISO and C9-ALS CS29) were co-stained with anti-TDP-43 antibody (red), anti-MAP2 antibody (magenta), and Hoechst 33258 (blue). The relative distribution of endogenous TDP-43 proteins was quantified similarly as above. Two-way ANOVA detected significant interaction effects of C9-ALS and lentiviral overexpression on the RAN gradient ($P$ = 0.0154 for LSM12; $P$ = 0.0120 for EPAC1). Data represent means ± SEM ($n$ = 102–104 GFP–positive cells from 4 independent differentiation experiments). n.s., not significant; $^*P < 0.05$, $^{**}P < 0.01$, $^{***}P < 0.001$, as determined by 2-way ANOVA with Tukey post hoc test. (G, H) Overexpression of LSM12 or EPAC1 suppresses caspase-3 activation in C9-ALS iPSNs. Three-week-old iPSNs (CS29-ISO and C9-ALS CS29) were co-stained with anti-cleaved caspase-3

antibody (red), anti-MAP2 antibody, and Hoechst 33258 (blue). The relative percentages of iPSNs expressing cleaved caspase-3 were averaged from 15 confocal images of random fields of interest per condition ($n$ = 771–1,129 GFP-positive cells from 3 independent differentiation experiments). Data represent means ± SEM. n.s., not significant; ***$P < 0.001$, as determined by 2-way ANOVA with Tukey post hoc test. All underlying numerical values are available in S1 Data. ANOVA, analysis of variance; ATXN2, ataxin-2; C9-ALS, C9ORF72-associated amyotrophic lateral sclerosis; EPAC1, exchange protein directly activated by cyclic AMP 1; GFP, green fluorescent protein; LSM12, like-Sm protein 12; N/C, nuclear to cytoplasmic; NCT, nucleocytoplasmic transport; NPC, neural progenitor cell; SEM, standard error of the mean.

we performed a series of immunoprecipitation (IP) experiments and compared the biochemical composition of RAN-containing protein complexes in different genetic settings and pharmacological conditions.

As expected, an anti-RANBP2 antibody co-purified RAN, importin β1, and SUMOylated RANGAP1, together with RANBP2, from control cell extracts (Fig 8A). *LSM12* deletion enhanced the association of SUMOylated RANGAP1 with RANBP2 while dissociating RAN and importin β1 from the RANBP2-RANGAP1 complex. Similar results were obtained by reciprocal IP using an anti-RANGAP1 antibody (S13A Fig). Pretreatment with a cell-permeable cAMP analog (8-pCPT-2-O-Me-cAMP-AM/007-AM) neither affected the assembly of wild-type RAN-associating protein complex nor rescued *LSM12*-deletion phenotypes (S13B Fig). EPAC1 activation by cAMP thus is unlikely involved in these biochemical interactions. Similar IP experiments in EPAC1-depleted cells further revealed that EPAC1 depletion caused dissociation of RAN and importin β1 from the RANBP2-RANGAP1 complex (S13C and S13D Fig), consistent with *LSM12*-deletion effects.

In either case, the enrichment of endogenous EPAC1 protein in RANBP2-RANGAP1 immunoprecipitates was not readily detectable, possibly owing to the abundance of free EPAC1 proteins in the cytoplasm compared with those associated with the RANBP2-RANGAP1 complex at the nuclear pore. Affinity purification of FLAG-tagged EPAC1 protein indeed confirmed its association with RANGAP1, RANBP2, importin β1, and RAN (Fig 8B). Moreover, overexpression of the FLAG-tagged EPAC1 protein restored the assembly of RAN-associating protein complexes in *LSM12*-deleted cells, consistent with its rescue effects on the RAN gradient and NCT. These results indicate that the *LSM12-EPAC1* pathway promotes the association of RAN and importin β1 with the RANBP2-RANGAP1 complex. Stronger interactions between RANBP2 and RANGAP1 in *LSM12*-deleted or EPAC1-depleted cells suggest their competitive interaction with RAN and importin β1.

## RAN overexpression mitigates EPAC1-depletion effects on poly(GR) toxicity

To determine whether the status of RAN-bound GTP hydrolysis contributes to *EPAC1* effects on the assembly of RAN-associating protein complexes, we added a hydrolysis-resistant GTP analog (GTPγS) to cell extracts before IP. The assembly of the RANBP2-associating protein complex was insensitive to GTPγS in control cells (Fig 8C). By contrast, preincubation of GTPγS with extracts from EPAC1-depleted cells restored the association of RAN and importin β1 with the RANBP2-RANGAP1 complex to control cell levels (Fig 8C). These results indicate that RAN-GTP is specifically limiting for the stable association of RAN-importin β1 with RANBP2-RANGAP1 in EPAC1-depleted cells. Given that the intracellular concentration of GTP is approximately 10-fold higher than that of GDP, we hypothesized that overexpression of wild-type RAN proteins would supply extra RAN-GTP in EPAC1-depleted cells and thereby rescue cellular phenotypes relevant to the lack of RAN-GTP. We indeed found that RAN overexpression substantially suppressed the formation of nuclear GR granules (Fig 8D) and poly(GR)-induced disruption of the nuclear membrane in EPAC1-depleted cells (Fig 8E and 8F), while modestly inhibiting poly(GR) effects in control cells. RAN overexpression also

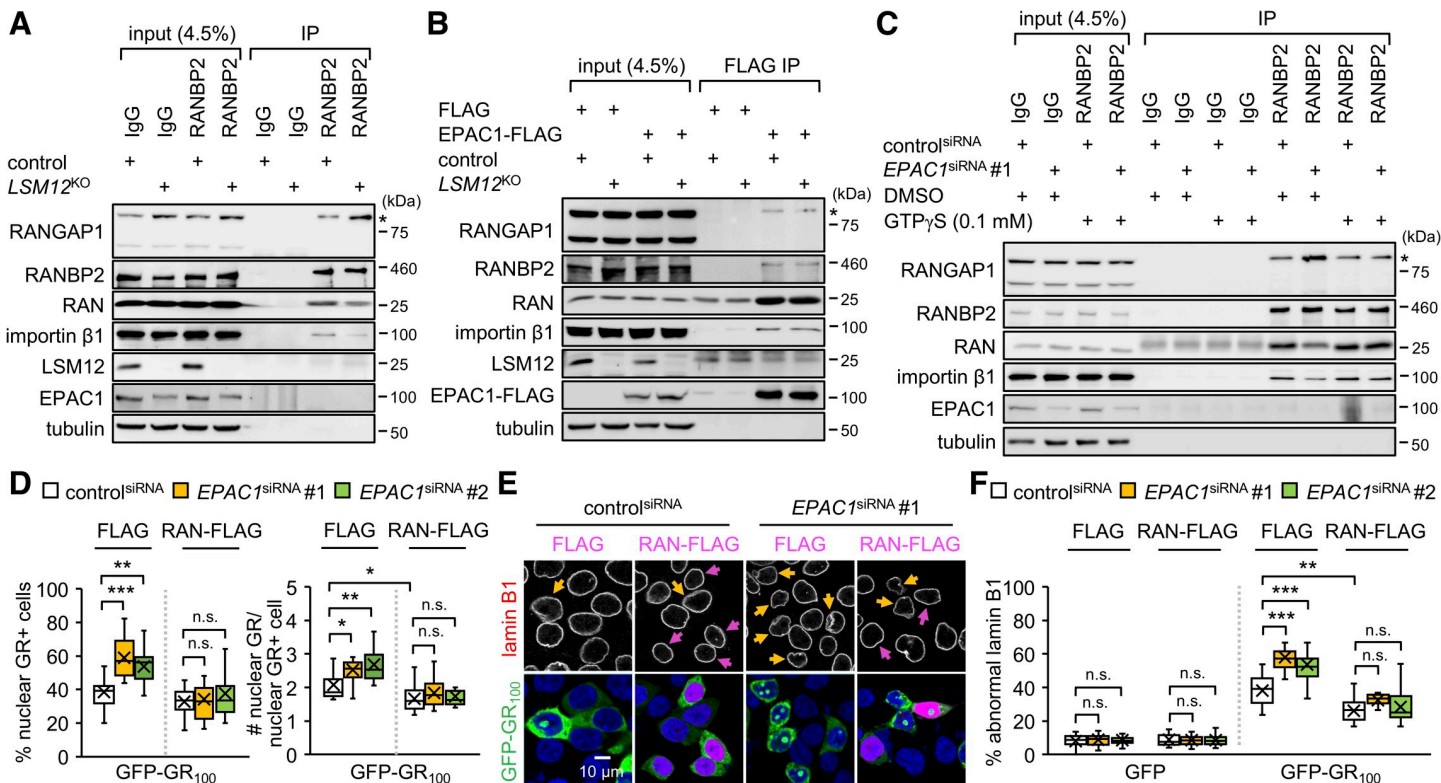

**Fig 8. EPAC1 depletion limits RAN-GTP availability for assembly of the RAN-associating nuclear pore complex and suppression of poly(GR) toxicity.** (A) *LSM12* deletion dissociates RAN and importin β1 from the RANBP2-RANGAP1 complex. Soluble extracts from control and *LSM12*^KO^ cells were immunoprecipitated with control IgG or anti-RANBP2 antibody. Purified IP complexes were resolved by SDS-PAGE and immunoblotted with specific antibodies (left). Asterisks indicate SUMOylated RANGAP1. Input, 4.5% of soluble extracts used in each IP. (B) EPAC1 overexpression restores the assembly of RAN-associating nuclear pore complex in *LSM12*-deleted cells. Control and *LSM12*^KO^ cells were transfected with FLAG or EPAC1-FLAG expression vector. Soluble extracts were prepared 48 hours after transfection and then immunoprecipitated with anti-FLAG antibody. (C) The hydrolysis-resistant GTP analog, GTPγS, blocks the dissociation of RAN and importin β1 from the RANBP2-RANGAP1 complex in *EPAC1*-depleted cells. Where indicated, soluble cell extracts were preincubated with 0.1 mM GTPγS or DMSO (vehicle control) at 25˚C for 30 minutes before IP. (D) RAN overexpression suppresses the nuclear assembly of poly(GR) granules. SH-SY5Y cells were co-transfected with siRNA, GFP-GR_{100}, and RAN-FLAG expression vectors, as in Fig 6A. Transfected cells were co-stained with anti-lamin B1 antibody (red), anti-FLAG antibody (magenta), and Hoechst 33258 (blue) 48 hours after plasmid DNA transfection. The assembly of nuclear poly(GR) granules was quantified similarly as in Fig 1. Data represent means ± SEM (*n* = 15–17 confocal images obtained from 3 independent experiments; *n* = 280–471 GFP-GR_{100}–positive cells). n.s., not significant; *P < 0.05, **P < 0.01, ***P < 0.001, as determined by 2-way ANOVA with Tukey post hoc test. (E, F) RAN overexpression suppresses the poly(GR)-induced disruption of the nuclear lamina. Yellow arrows indicate GFP-GR_{100}–positive cells with severe nuclear lamina disruption. Magenta arrows indicate cells overexpressing RAN-FLAG protein. The abnormal morphology of the nuclear lamina was quantified as in Fig 3E. Data represent means ± SEM (*n* = 16–17 confocal images obtained from 3 independent experiments; *n* = 325–547 GFP–or GFP-GR_{100}–positive cells). n.s., not significant; **P < 0.01, ***P < 0.001, as determined by 2-way ANOVA with Tukey post hoc test. All underlying numerical values are available in S1 Data. ANOVA, analysis of variance; EPAC1, exchange protein directly activated by cyclic AMP 1; GFP, green fluorescent protein; IP, immunoprecipitation; LSM12, like-Sm protein 12; SEM, standard error of the mean; siRNA, small interfering RNA.

suppressed EPAC1-depletion effects on poly(GR)-induced disruption of NCT, although it had negligible effects in control cells (S14A and S14B Fig). Taken together, these data demonstrate that the *LSM12-EPAC1* pathway facilitates the nucleocytoplasmic recycling of RAN-GTP and thereby confers cellular resistance to poly(GR)-related pathogenic processes.

## Discussion

Identification of noncanonical translation products from hexanucleotide repeat expansions in the *C9ORF72* locus and their impact on specific aspects of cell physiology have advanced our understanding of the pathogenesis of ALS/FTD [22,23]. *C9ORF72*-derived poly(GR) proteins assemble into distinct intracellular compartments, yet they also induce the formation of SGs [94]. Nonfunctional sequestration of NCT-related factors into SGs has been proposed as a key

mechanism for the NCT deficits implicated in ALS/FTD [33]. This has been validated by modifier screens in yeast and *Drosophila* genetic models [37,39,95], indicating strong conservation of their pathogenic mechanism. Mutations in *ATXN2*, one of these genetic modifiers, are indeed associated with ALS [42], and it has been shown that *ATXN2* facilitates neurodegeneration, in part, by promoting SG formation [33,43]. As expected, we found that ATXN2 and ATXN2-associated LSM12 make a nonadditive contribution to the dynamic assembly of SGs. Unexpectedly, however, we discovered opposing effects of *LSM12* and its posttranscriptional downstream effector, *EPAC1*, on neurodegeneration. The *LSM12-EPAC1* pathway assembles the RAN-associating nuclear pore complex and establishes the nucleocytoplasmic RAN gradient, thereby antagonizing the effects of poly(GR) protein on NCT and nuclear membrane integrity. Our definition of the neuroprotective *LSM12-EPAC1* pathway was further supported by its suppression of NCT-relevant pathologies and caspase-3 activation for apoptosis in C9-ALS iPSNs.

We propose that EPAC1 mediates the immediate recycling of nuclear RAN-GTP by supporting the stable association of RAN-importin β1 with RANBP2-RANGAP1 at the cytoplasmic side of the nuclear pore complex (S15 Fig). An EPAC1 deficiency, on the other hand, would allow nuclear RAN-GTP to diffuse out of the nuclear pore and further localize in the cytoplasm. The cytoplasmic RANGAP complex (e.g., RANBP1-RANGAP) then mediates RAN-GTP hydrolysis [96], thereby delaying the nuclear import of RAN-GDP by nuclear transport factor 2 (NTF2) [97–99] and limiting the regeneration of RAN-GTP from RAN-GDP by nuclear RCC1/RANGEF. RAN overexpression may increase the total flux of RAN recycling and elevate the local concentration of RAN at the nuclear pore complex. Consequently, RAN overexpression could facilitate NTF2-dependent nuclear import of RAN-GDP in EPAC1-depleted cells, partially restore nucleocytoplasmic RAN gradient, and compensate for the loss of EPAC1 function in the assembly of RAN-associating protein complexes at the nuclear pore. This model may explain our observation that RAN overexpression rescues EPAC1-depletion phenotypes in NCT and poly(GR) toxicity. Notably, RAN overexpression similarly suppresses the cytotoxic effects of mutant huntingtin through a mechanism that also involves disruption of the RAN gradient and NCT [72].

When transiently expressed in cell culture, poly(GR) proteins display heterogeneity in their subcellular expression patterns, ranging from a uniform distribution in the cytoplasm to nucleolar localization [25,30,33]. We found that cytoplasmic poly(GR) protein accumulated gradually in subnuclear compartments, a localization pattern that strongly correlated with the loss of nuclear integrity, as assessed by invaginations of the nuclear lamina. Given the active role of RAN in nucleating formation of the nuclear envelope [100,101] and assembly of the nuclear pore complex [102,103], we speculate that poly(GR)-induced disruption of the RAN gradient feeds forward to increase the permeability of the nuclear envelope, allowing more cytoplasmic poly(GR) protein to enter the nucleus and trigger nuclear pathogenesis, including nucleolar stress [25,26,30,31]. Consistent with this idea, we showed that RAN overexpression was sufficient to suppress the nuclear translocation of cytoplasmic poly(GR) proteins and sustain nuclear integrity in poly(GR)-expressing cells. This aspect of RAN function is relevant to the chromatin association of RAN and RCC1/RANGEF, which intrinsically cues the relative location of nuclei and generates the RAN-GTP gradient during mitosis [104,105]. Nonetheless, it remains to be determined if poly(GR)-induced disruption of the RAN gradient comparably affects the functionality of the nuclear envelope in postmitotic cells since some conflicting observations were made on the nuclear envelope invaginations between C9-ALS iPSNs and postmortem cortical tissue from *C9ORF72*-associated ALS patients [106].

Genetic mutations in nuclear lamin genes cause a group of rare genetic disorders, collectively called laminopathies [107]. For instance, premature aging in Hutchinson–Gilford

progeria syndrome (HGPS), caused by a lamin A/C mutation, manifests as cytological defects in the nuclear envelope and heterochromatin formation [108–110]. Lamin A/C mutations, including the pathogenic allele observed in HGPS, lead to disruption of the RAN gradient and NCT [111–114]. These mutant phenotypes appear to be mediated by loss of heterochromatin since genetic or pharmacological inhibition of heterochromatin formation by histone methyltransferases is sufficient to disrupt the RAN gradient and NCT [115]. Given our observation that poly(GR)-induced loss of nuclear integrity is rescued by RAN overexpression, these findings indicate possible interdependence among nuclear envelop integrity, heterochromatin formation, and the RAN gradient. Interestingly, loss of heterochromatin formation and nuclear lamina invaginations have been implicated in ALS/FTD-related pathogenic processes in a mouse model [116]. An additional layer of homology is likely present between HGPS and ALS/FTD, given that deficits in the nuclear import of DNA-repairing factors (e.g., ataxia-telangiectasia mutated kinase) may explain their common pathogenesis, which involves DNA damage-induced cell death [89,115,117–119].

The nuclear pore complex is exceptionally long lived in postmitotic cells [120,121], and its relevant function, such as NCT and nuclear integrity, declines with aging [110,122,123]. Emerging evidence indicates that cellular pathogeneses underlying distinct neurodegenerative diseases, including ALS/FTD, Alzheimer's disease, and Huntington's disease, may converge on the disruption of NCT [71–73,124,125]. Given our findings that EPAC1 acts as a gatekeeper in the vicinity of nuclear pores to facilitate RAN recycling and sustain robust NCT, the posttranscriptional circuit of *LSM12-EPAC1* would enrich the repertoire of antiaging molecular pathways that have evolved on fundamental cellular physiology.

## Materials and methods

### Cell culture

Human neuroblastoma SH-SY5Y and human embryonic kidney 293T cells were cultured in Dulbecco's Modified Eagle Medium (HyClone, Logan, Utah, United States of America) supplemented with 10% fetal bovine serum and 1% antibiotics and maintained at 37˚C in a humidified incubator with 95% air and 5% $CO_2$. Plasmid DNA and siRNA were transiently transfected using polyethylenimine [126] and Lipofectamine RNAiMAX transfection reagent (Thermo Fisher Scientific, Waltham, Massachusetts, USA), respectively, according to the manufacturer's instructions. C9-ALS iPSC lines (CS28iALS, CS29iALS, and CS52iALS) and control iPSC lines (CS0YX7iCTR, CS29iALS-ISO, and CS4NWCiCTR) were obtained from the Cedars-Sinai iPSC Core. iPSCs were cultured on Matrigel-coated plates (Corning, Corning, New York, USA) in mTeSR1 medium (STEMCELL Technologies, Vancouver, British Columbia, Canada) in a humidified incubator with 95% air and 5% $CO_2$ with daily media changes. Motor neuron differentiation from iPSCs was induced, as described previously [81]. Where indicated, neural progenitor cells (NPCs) were transduced with concentrated aliquots of recombinant lentiviruses in the presence of 5 μg/ml polybrene for 6 hours to overexpress LSM12 or EPAC1. iPSNs were harvested 21 days after neuronal differentiation from NPC and subsequently analyzed.

### *Drosophila* melanogaster

Flies were raised on standard cornmeal–yeast–agar medium (5.4% cornmeal, 1.3% yeast, 0.7% soy flour, 0.4% agar, 4.2% starch syrup, 0.4% propionic acid, and 0.8% methyl 4-hydroxybenzoate) at 25˚C. UAS-poly-GR.PO-36 (BL58692), *Epac* deletion mutant ($Epac^{\Delta 1}/Epac^{\Delta 3}$, BL78799), and UAS-*Epac* RNAi (v43444, v110077) lines were obtained from the Bloomington *Drosophila* Stock Center and Vienna *Drosophila* Resource Center.

## DNA constructs

The nucleotide sequences for *LSM12* and *ATXN2* shRNAs were selected from a predesigned library (Dharmacon, Lafayette, Colorado, USA). A control shRNA harbored a nucleotide sequence (5′-TCC TAA GGT TAA GTC GCC CTC-3′) that did not target any mammalian transcripts. Oligonucleotides encoding each shRNA were synthesized (Macrogen, Seoul, Korea) and subcloned into pLKO.1 (a gift from B. Weinberg; Addgene plasmid #8453) for the production of recombinant lentiviruses. The nucleotide sequences for *EPAC1* siRNAs were similarly selected from the predesigned library (Dharmacon). *EPAC1* siRNAs and non-targeting control siRNA (5′-CCU CGU GCC GUU CCA UCA GGU AGU U-3′) were synthesized (Genolution, Seoul, Korea), dissolved in PBS, and stored at −20°C before use. A small guide RNA (sgRNA) sequence (5′-GAC GTG AGT TGG GAT CGG AG-3′) for the CRISPR/Cas9-mediated deletion of the *LSM12* locus was designed using CHOPCHOP (https://chopchop.rc.fas.harvard.edu). The oligonucleotide encoding *LSM12* sgRNA was synthesized (Macrogen) and subcloned into pRGEN-U6-sgRNA and pHRS (ToolGen, Seoul, Korea) for the expression of *LSM12* sgRNA and the hygromycin B-resistance reporter gene, respectively. Full-length cDNAs encoding ATXN2, EPAC1, LSM12, or RAN proteins were PCR-amplified from pcDNA6-ATXN2-22Q (a gift from A. Gitler), pEYFP-N3-EPAC1 (a gift from X. Cheng; Addgene plasmid #113110) or SH-SY5Y cDNA samples, respectively, and then subcloned into a modified pcDNA3.1 (pcDNA-3xnF) for the expression of carboxyl-terminal triple-FLAG fusion proteins. Poly(GR)-encoding cDNA (a gift from D. Ito) was subcloned into a modified pcDNA3.1 (pcDNA-GFP) vector for the expression of N-terminal GFP fusion proteins. RPL10A-encoding cDNA was PCR-amplified from SH-SY5Y cDNA samples and subcloned into a modified pcDNA-GFP vector containing a hygromycin B-resistance gene for the selection of cells stably expressing N-terminally GFP-fused RPL10A proteins. Subregions of the *EPAC1* locus were PCR-amplified from SH-SY5Y genomic DNA samples. The DNA fragments corresponding to *EPAC1* 5′ UTR or 3′ UTR were subcloned into a modified pcDNA 3.1 (pcDNA-NLUC) vector for the expression of NLUC-encoding UTR reporter transcripts. The DNA fragment corresponding to an *EPAC1* promoter region was subcloned into a modified pGL2 basic (pGL2-NLUC) vector for the NLUC reporter expression by the *EPAC1* promoter activity. FLUC-encoding cDNA was subcloned into pcDNA3.1 for the expression of a normalizing control reporter along with the NLUC reporters. S-GFP was a gift from M.S. Hipp [64]. S-tdT-encoding cDNA (a gift from J. D. Rothstein) was subcloned into pcDNA3.1. Lentiviral vectors for overexpression of LSM12 or EPAC1 were constructed by the Gateway cloning pENTR-*Ubiquitin C* promoter (Addgene plasmid #45959) and modified pENTR4-FLAG entry vector (Addgene plasmid #17424) harboring either LSM12 or EPAC1 cDNA into pCWX-DEST destination vector (Addgene plasmid #45957).

## Production of recombinant lentiviruses and establishment of stable cell lines

The 293T cells were co-transfected with viral packaging plasmids (ViraPower Lentiviral Packaging Mix; Invitrogen, Carlsbad, California, USA) and third-generation pLKO.1 lentiviral vector encoding control, *LSM12*, or *ATXN2* shRNA. Cell culture medium containing recombinant lentiviruses was harvested 48 and 72 hours after transfection. SH-SY5Y cells were then incubated with the lentiviral medium for 48 hours. Stably infected cells were selected by replacing fresh media containing 1 μg/ml puromycin every 3 days for 2 weeks to establish a stable cell line. Lentiviral particles for LSM12 or EPAC1 overexpression were produced similarly as above and concentrated by ultracentrifugation at 25,000 rpm for 2 hours at 4°C.

## Establishment of an *LSM12*-knockout (*LSM12*$^{\text{KO}}$) cell line by CRISPR/Cas9-mediated deletion

SH-SY5Y cells were co-transfected with pRGEN-Cas9-CMV, pRGEN-U6-*LSM12* sgRNA, and pHRS-*LSM12* sgRNA reporter. Reporter-edited cells were selected by culturing in medium containing 500 μg/ml hygromycin B for 2 weeks. Individual colonies were manually picked and seeded into 12-well plates containing hygromycin B-free media to establish independent cell lines. LSM12 expression in each cell line was analyzed by immunoblotting total cell extracts with an anti-LSM12 antibody. Cell lines lacking detectable LSM12 expression were further analyzed, whereas those showing LSM12 expression comparable to that of parental cells served as controls.

## Immunofluorescence assay

SH-SY5Y cells were grown on coverslips and then fixed in phosphate-buffered saline (PBS) containing 3.7% formaldehyde at 25°C for 15 minutes. NPCs were seeded on Matrigel-coated coverslips. Twenty-one days after neuronal differentiation from NPC, iPSNs were fixed similarly as above. After 2 washes with PBS, fixed cells were permeabilized with PBS containing 0.1% Triton X-100 (PBS-T) at 4°C for 15 minutes, followed by 2 washes with PBS. Permeabilized cells were blocked with PBS-T containing 1% bovine serum albumin at 25°C for 30 minutes and then incubated with primary antibodies diluted in blocking buffer containing 0.05% sodium azide at 4°C for 1 day. The primary antibodies used in immunostaining were mouse anti-RAN (1:1,000; Santa Cruz Biotechnology, Santa Cruz, California, USA), mouse anti-G3BP1 (1:2,000; Santa Cruz Biotechnology), mouse anti-FLAG (1:2,000; Sigma-Aldrich, St. Louis, Missouri, USA), rabbit anti-lamin B1 (1:1,000; Proteintech, Chicago, Illinois, USA), rabbit anti-G3BP1 (1:2,000; Proteintech), mouse anti-PABPC1 (1:2,000; Santa Cruz Biotechnology), rabbit anti-ATXN2 (1:2,000; Proteintech), guinea pig anti-MAP2 (1:1,000; Synaptic System, Goettingen, Germany), mouse anti-HB9 (1:1,000; Developmental Studies Hybridoma Bank, Iowa City, Iowa, USA), and rabbit anti-ChAT (1:1,000; Proteintech), rabbit anti-polyGR (1:250; Proteintech), rabbit anti-TDP43 (1:1,000; Proteintech), and rabbit anti-cleaved caspase-3 (1:400; Cell Signaling Technology, Beverly, Massachusetts, USA). After incubation with primary antibodies, cells were washed twice with PBS-T for 5 minutes each and then incubated at 4°C for 1 day with secondary antibodies diluted 1:600 in PBS-T. Species-specific Alexa Flour 488-, 594-, or 647-conjugated anti-IgG antibodies were used as secondary antibodies for immunostaining (Jackson ImmunoResearch Laboratories, West Grove, Pennsylvania, USA). Nuclei were visualized by staining with Hoechst for 5 minutes, followed by washing twice with PBS-T. Stained samples were slide-mounted in VECTASHIELD antifade mounting medium (Vector Laboratories, Burlingame, California, USA).

## Quantitative image analysis

Several fields of interest were randomly selected from each imaging sample, and their confocal images were individually obtained using an FV1000 microscope (Olympus, Tokyo, Japan) with identical imaging settings. The fluorescence intensities of confocal images were quantified using ImageJ software. For SG analysis, G3BP-positive cytoplasmic inclusions with fluorescence intensities above a threshold were unbiasedly scored as SGs using ImageJ. The total numbers of cells, SG-positive cells and SGs per field, as well as the diameter of individual SGs, were measured as described previously [127]. For nuclear poly(GR) granule analysis, the percentage of cells positive for nuclear poly(GR) granules per field and the number of nuclear poly(GR) granules per cell were manually scored. The relative distribution of S-GFP, S-tdT,

TDP-43, or RAN protein between nucleus and cytoplasm (N/C ratio) was quantified by calculating the ratio of nuclear to cytoplasmic fluorescence from each reporter protein, anti-TDP-43 or anti-RAN antibody staining per cell, as described previously [64,128].

## Quantitative RNA analysis

Total RNA was purified using the TRIzol reagent (Thermo Fisher Scientific) and digested with DNase I according to the manufacturer's instructions (Promega, Madison, Wisconsin, USA). DNase I was subsequently removed by phenol–chloroform extraction. RNA samples were further purified by ethanol precipitation and reverse-transcribed using M-MLV reverse transcriptase (Promega) with random hexamers. cDNA samples were quantitatively analyzed using SYBR Green-based Prime Q-Mastermix (GeNet Bio, Nonsan, Korea) and gene-specific primers on a LightCycler 480 real-time PCR system (Roche, Basel, Switzerland).

## Luciferase reporter assay

SH-SY5Y cells on 12-well plates were co-transfected with 100 ng of *EPAC1* NLUC reporter plasmid and 100 ng of FLUC control plasmid. Transfected cells were harvested 48 hours after transfection and lysed for luciferase assays using a Nano-Glo Dual Luciferase Assay kit (Promega) according to the manufacturer's instructions. Luminescence from each cell lysate was measured using a GloMax Navigator microplate luminometer (Promega).

## Translating ribosome affinity purification (TRAP)

TRAP was performed as described previously [129] with minor modifications. Briefly, SH-SY5Y cells stably expressing GFP-RPL10A were collected and lysed in a lysis buffer (20 mM HEPES-KOH pH 7.4, 150 mM KCl, 10 mM MgCl$_2$, 1%(v/v) NP-40, 100 μg/ml cycloheximide, 1 mM dithiothreitol [DTT], 1 mM phenylmethylsulfonyl fluoride [PMSF], and 200 U/ml RNase inhibitor) at 4°C for 15 minutes with gentle rocking. Lysates were first centrifuged at 2,000 × g for 5 minutes at 4°C, and the supernatant was collected and further clarified by centrifugation at 20,000 × g for 10 minutes at 4°C. After clarification by centrifugation, 30 μl of soluble extracts were retained as input, and the remaining soluble extract was incubated with anti-GFP antibody (NeuroMab, Davis, California, USA) at 4°C for 1.5 hours with gentle rocking. Pre-equilibrated Protein G magnetic beads (New England BioLabs, Ipswich, Massachusetts, USA) were added to the extracts and further incubated at 4°C for 1.5 hours with gentle rocking. The beads were washed 4 times with a wash buffer (20 mM HEPES-KOH pH 7.4, 350 mM KCl, 5 mM MgCl$_2$, 1%(v/v) NP-40, 100 μg/ml cycloheximide, 1 mM DTT, 1 mM PMSF, and 200 U/ml RNase inhibitor) prior to RNA extraction. Input and translating ribosome-associated RNAs were purified using an RNeasy Mini kit (QIAGEN, Hilden, Germany) according to the manufacturer's instructions.

## Differential gene expression analysis by RNA sequencing

Ribosomal RNA was depleted from 50 ng of purified RNA samples using an rRNA depletion kit (New England BioLabs). cDNA libraries for RNA sequencing were generated using a NEBNext Ultra Directional RNA Library Prep Kit for Illumina according to the manufacturer's instructions (New England BioLabs). The quality of sequencing libraries, defined by RNA integrity number (RIN) values, was examined using a DNA1000 chip (Agilent, Santa Clara, California, USA) and Agilent Technologies 2100 Bioanalyzer. After quantification by qRT-PCR using SYBR Green PCR Master Mix (Applied Biosystems, Foster City, California, USA), an equimolar amount of index-tagged libraries was pooled into a single sample. Clusters

were generated in the flow cell on the cBot automated cluster-generation system (Illumina, San Diego, California, USA). RNA sequencing was performed on a NovaSeq 6000 system (Illumina) with a 2× 100-bp read length (DNA Link, Seoul, Korea). Sequence reads were first mapped to the reference genome (Human hg19) using Tophat (v2.0.13) [130,131], yielding 50 to 75 million mapped reads per sample. The aligned results were added to Cuffdiff (v2.2.0), and differentially expressed genes (DEGs) were identified using the Cuffdiff output file, "gene_exp.diff."

## Immunoprecipitation (IP)

SH-SY5Y cells on a 100-mm dish were collected and lysed in a lysis buffer (25 mM Tris-Cl pH 7.5, 150 mM NaCl, 10% glycerol, 1 mM EDTA, 1 mM DTT, 0.5%(v/v) NP-40, and 1 mM PMSF) at 4˚C for 15 minutes with gentle rocking. After clarification by centrifugation, 30 μl of soluble extracts were retained as input, and the remaining soluble extract was incubated with 2 μg of antibody or control IgG at 4˚C for 1.5 hours with gentle rocking. Pre-equilibrated Protein G magnetic beads (New England BioLabs) were added to the extracts and further incubated at 4˚C for 1.5 hours with gentle rocking. The beads were then washed 5 times in the same buffer. Input samples and immunoprecipitates were resolved by SDS-PAGE on 6.5% and 10% gels, and then proteins in each gel were transferred to Protran nitrocellulose membranes (GE Healthcare, Chicago, Illinois, USA) and incubated with a specific set of primary antibodies. The primary antibodies used in immunoblotting were mouse anti-RAN (1:1,000; Santa Cruz Biotechnology), mouse anti-importin β1 (1:1,000; Santa Cruz Biotechnology), mouse anti-RANGAP1 (1:1,000; Santa Cruz Biotechnology), mouse anti-RANBP2 (1:1,000; Santa Cruz Biotechnology), mouse anti-EPAC1 (1:500; Santa Cruz Biotechnology), rabbit anti-LSM12 (1:2,000), rabbit anti-p-EIF2α (1:2,000; Cell signaling), mouse anti-EIF2α (1:1,000; Santa Cruz Biotechnology), mouse anti-tubulin (1:1,000; Developmental Studies Hybridoma Bank), rabbit anti-ATXN2 (1:2,000; Proteintech), rabbit anti-ATXN2L (1:2,000; Medico, Ansan, Korea), mouse anti-PABPC1 (1:1,000; Santa Cruz Biotechnology), and mouse anti-FLAG (1:2,000; Sigma-Aldrich). Immunoreactive proteins were recognized by species-specific horseradish peroxidase-conjugated secondary antibodies (Jackson Immuno Research Laboratories), with subsequent detection by Clarity Western ECL blotting substrate (Bio-Rad, Hercules, California, USA) using ImageQuant LAS 4000 (GE Healthcare).

## Quantification and statistical analysis

Quantification procedures for all experimental analyses are described in figure legends and Materials and Methods above. Microsoft Excel and GraphPad Prism 8 software were used for Student $t$ test, 1-way analysis of variance (ANOVA), and 2-way ANOVA with post hoc tests. The statistical details of experiments, including the number of experiments and samples analyzed, the statistical tests used, and $P$ values, are described in the figure legends.

## Supporting information

**S1 Fig. LSM12 depletion impairs SG assembly under arsenite-induced oxidative stress.** (A) Puromycin treatment enhances arsenite-induced SG formation in control[shRNA] cells, but not in *LSM12*[shRNA] cells. Control[shRNA] and *LSM12*[shRNA] cells were preincubated with puromycin (5 μg/ml) or PBS (vehicle control) for 15 minutes before the induction of chronic oxidative stress (50-μM NaAsO$_2$, 2 hours). Cells were co-stained with anti-G3BP1 antibody (red), anti-PABPC1 antibody (green), and Hoechst 33258 (blue) to visualize SGs and the nucleus, respectively. The percentage of SG-positive cells, the number of SGs per cell, and the size of SGs were quantified using ImageJ software and averaged ($n$ = 10 confocal images obtained from 3

independent experiments; $n$ = 713–939 cells). Error bars indicate SEM. n.s., not significant; *$P$ < 0.05, **$P$ < 0.01, ***$P$ < 0.001, as determined by 2-way ANOVA with Tukey post hoc test. (B) LSM12 depletion altered the kinetics of SG disassembly during the recovery from acute oxidative stress. Control$^{shRNA}$ and *LSM12*$^{shRNA}$ cells were incubated with 500-μM NaAsO$_2$ for 1 hour and then washed with fresh media. Cells were fixed at the indicated time after the removal of NaAsO$_2$ and co-stained with anti-G3BP1 antibody (red), anti-PABPC1 antibody (green), and Hoechst 33258 (blue). (C) The kinetics of SG disassembly were quantified similarly as above. Data represent means ± SEM ($n$ = 8–10 confocal images obtained from 3 independent experiments; $n$ = 388–740 cells). *$P$ < 0.05, **$P$ < 0.01, ***$P$ < 0.001, as determined by Student $t$ test. All underlying numerical values are available in S1 Data. ANOVA, analysis of variance; LSM12, like-Sm protein 12; SG, stress granule; SEM, standard error of the mean.
(TIFF)

**S2 Fig. Decreased phosphorylation of EIF2α underlies the impairment of arsenite-induced SG formation in LSM12-depleted cells.** (A) LSM12 depletion does not suppress the SG assembly under sorbitol-induced osmotic stress. Control$^{shRNA}$ and *LSM12*$^{shRNA}$ cells were incubated with 0.4-M sorbitol for the indicated time before co-staining with anti-ATXN2 antibody (red), anti-G3BP1 antibody (green), and Hoechst 33258 (blue). (B, C) LSM12 depletion decreases the arsenite-induced phosphorylation of EIF2α. Control$^{shRNA}$ and *LSM12*$^{shRNA}$ cells were incubated with 50-μM NaAsO$_2$, 1-μM thapsigargin (Tg), or 0.4-M sorbitol for the indicated time before harvest. Whole-cell lysates were immunoblotted with specific antibodies (left). Relative levels of EIF2α phosphorylation were calculated by normalizing the ratio of phospho-EIF2α to EIF2α protein levels per condition to that in control cells with no chemical treatment. Data represent means ± SEM ($n$ = 3). n.s., not significant; *$P$ < 0.05, **$P$ < 0.01, as determined by Student $t$ test. All underlying numerical values are available in S1 Data. ATXN2, ataxin-2; EIF2α, eukaryotic translation initiation factor 2 subunit α; LSM12, like-Sm protein 12; SEM, standard error of the mean; SG, stress granule.
(TIFF)

**S3 Fig. LSM12 depletion exacerbates the impairment of NCT caused by arsenite-induced oxidative stress.** (A) LSM12 depletion facilitates the nuclear mislocalization of S-GFP under oxidative stress conditions. Control$^{shRNA}$ and *LSM12*$^{shRNA}$ cells were transfected with an S-GFP expression vector and then treated with 50-μM NaAsO$_2$ for the indicated time before co-staining with anti-G3BP1 antibody (red) and Hoechst 33258 (blue). NCT of S-GFP reporter proteins was quantified as in Fig 2B. Data represent means ± SEM ($n$ = 123–127 cells from 3 independent experiments). *$P$ < 0.05, **$P$ < 0.01, ***$P$ < 0.001 to control$^{shRNA}$ cells at a given time point, as determined by Student $t$ test. (B) LSM12 depletion facilitates the cytoplasmic mislocalization of S-tdT under oxidative stress conditions. Data represent means ± SEM ($n$ = 103–118 cells from 3 independent experiments). n.s., not significant; *$P$ < 0.05, **$P$ < 0.01 to control$^{shRNA}$ cells at a given time point, as determined by Student $t$ test. All underlying numerical values are available in S1 Data. GFP, green fluorescent protein; LSM12, like-Sm protein 12; NCT, nucleocytoplasmic transport; S-tdT, S-tdTomato; SEM, standard error of the mean.
(TIFF)

**S4 Fig. *LSM12* deletion increases the toxicity of *C9ORF72*-derived poly(GR) protein.** (A) *LSM12* deletion exacerbates the poly(GR)-induced invaginations of the nuclear envelope. Control and *LSM12*$^{KO}$ cells were transfected with a GFP-GR$_{100}$ expression vector and then co-stained with anti-lamin B1 antibody (red) and Hoechst 33258 (blue) 48 hours after

transfection. Yellow arrows indicate GFP-GR$_{100}$–positive cells with severe nuclear lamina disruption. (B) The assembly of nuclear poly(GR) granules was quantified as in Fig 1. Data represent mean ± SEM ($n$ = 15–17 confocal images obtained from 3 independent experiments; $n$ = 403–418 GFP-GR$_{100}$–positive cells). $^{***}P < 0.001$, as determined by Student $t$ test. (C) The abnormal morphology of the nuclear lamina was quantified as in Fig 3E. Data represent means ± SEM ($n$ = 18–19 confocal images obtained from 3 independent experiments; $n$ = 366–413 GFP–or GFP-GR$_{100}$–positive cells). n.s., not significant; $^{***}P < 0.001$, as determined by 2-way ANOVA with Tukey post hoc test. All underlying numerical values are available in S1 Data. ANOVA, analysis of variance; GFP, green fluorescent protein; LSM12, like-Sm protein 12; SEM, standard error of the mean.
(TIFF)

**S5 Fig. LSM12$^{V135I}$ overexpression does not alter the relative levels of endogenous LSM12 protein.** (A) SH-SY5Y cells were transfected with an expression vector for FLAG, LSM12-FLAG, or LSM12$^{V135I}$-FLAG. Total cell extracts were prepared 48 hours after transfection and immunoblotted with anti-FLAG, anti-LSM12, anti-ATXN2, anti-PABPC1, and anti-tubulin (loading control) antibodies. Overexpression of wild-type LSM12, but not LSM12$^{V135I}$, increased the relative levels of endogenous ATXN2 protein, consistent with low levels of endogenous ATXN2 protein in LSM12-depleted cells (Fig 1C). (B) The abundance of each protein was quantified as in Fig 1C. Data represent means ± SEM ($n$ = 4). n.s., not significant; $^{***}P < 0.001$, as determined by 1-way ANOVA with Dunnett post hoc test. All underlying numerical values are available in S1 Data. ANOVA, analysis of variance; ATXN2, ataxin-2; LSM12, like-Sm protein 12; SEM, standard error of the mean.
(TIFF)

**S6 Fig. Quantitative analyses of total mRNAs (RNA-seq) and translating ribosome-associated mRNAs (TRAP-seq) are reproducible between 2 biological replicates of control$^{shRNA}$ and *LSM12*$^{shRNA}$ cells.** The scatter plots of RNA-seq and TRAP-seq analyses were obtained from 2 biological replicates of control$^{shRNA}$ and *LSM12*$^{shRNA}$ cells. Pearson correlation and $P$ values are indicated in each plot. All underlying numerical values are available in S1 Data. LSM12, like-Sm protein 12; RNA-seq, RNA sequencing; TRAP-seq, translating ribosome affinity purification sequencing.
(TIFF)

**S7 Fig. *LSM12* deletion posttranscriptionally down-regulates *EPAC1* expression.** (A) *LSM12*-deleted (*LSM12*$^{KO}$) cells express low levels of EPAC1 protein. The abundance of each protein was quantified as in Fig 1C. Data represent means ± SEM ($n$ = 3). $^{**}P < 0.01$, as determined by Student $t$ test. (B) A schematic representation of the *EPAC1* locus and *EPAC1* reporter constructs. Transcription of control and *EPAC1* UTR reporters was driven by heterologous CMV promoter. A promoter region in the *EPAC1* locus (from −1,805 to +71 in relative to the transcription start site +1) was subcloned upstream of the NLUC-coding sequence to measure the *EPAC1* promoter activity by the NLUC activity. (C) *LSM12* deletion posttranscriptionally decreases *EPAC1* expression via the 5′ UTR. Control and *LSM12*$^{KO}$ cells were cotransfected with each *EPAC1* reporter and a FLUC expression vector (normalizing control). Luciferase reporter assays were performed as in Fig 5D. Data represent means ± SEM ($n$ = 3). n.s., not significant; $^{*}P < 0.05$, $^{**}P < 0.01$ as determined by Student $t$ test. All underlying numerical values are available in S1 Data. CMV, cytomegalovirus; EPAC1, exchange protein directly activated by cyclic AMP 1; FLUC, firefly luciferase; LSM12, like-Sm protein 12; NLUC, Nano-luciferase; SEM, standard error of the mean; UTR, untranslated region.
(TIFF)

**S8 Fig. EPAC1 depletion is sufficient to phenocopy loss of *LSM12* function in poly(GR) toxicity.** (A) SH-SY5Y cells were transfected with control or 2 independent *EPAC1* siRNAs. Total cell extracts were prepared 72 hours after transfection and immunoblotted with anti-EPAC1 or anti-tubulin (loading control) antibodies. (B) *LSM12* deletion and EPAC1 depletion nonadditively disrupt the RAN gradient. Control and $LSM12^{KO}$ cells were transfected with $control^{siRNA}$ or $EPAC1^{siRNA}$. The nucleocytoplasmic RAN gradient was quantified as in Fig 2D. Data represent means ± SEM ($n$ = 62–67 cells from 3 independent experiments). n.s., not significant; $^{***}P < 0.001$, as determined by 2-way ANOVA with Tukey post hoc test. (C, D) EPAC1 depletion exacerbates the poly(GR)-induced disruption of NCT in an ISRIB-insensitive manner. SH-SY5Y cells were co-transfected with siRNA and expression vectors for S-tdT and GFP-$GR_{100}$, as in Fig 6A. Where indicated, transfected cells were incubated with 2-μM ISRIB or DMSO (vehicle control) for 5 hours before co-staining with anti-G3BP1 antibody (magenta) and Hoechst 33258 (blue) at 48 hours after plasmid DNA transfection. NCT of S-tdT reporter proteins was quantified as in Fig 2B. Two-way ANOVA detected significant interaction effects of GFP-$GR_{100}$ and ISRIB treatment on NCT in $control^{siRNA}$ cells ($P = 0.0097$), but not in $EPAC1^{siRNA}$ cells ($P = 0.5310$ for $EPAC1^{siRNA}$ #1; $P = 0.5218$ for $EPAC1^{siRNA}$ #2); significant interaction effects of GFP-$GR_{100}$ and EPAC1 depletion on NCT regardless of ISRIB treatment ($P = 0.0194$ for $EPAC1^{siRNA}$ #1 in DMSO; $P = 0.0133$ for $EPAC1^{siRNA}$ #2 in DMSO; $P < 0.0001$ for both $EPAC1^{siRNA}$ in ISRIB). Data represent means ± SEM ($n$ = 128–140 GFP– or GFP-$GR_{100}$–positive cells from 3 independent experiments). $^{*}P < 0.05$, $^{**}P < 0.01$, $^{***}P < 0.001$, as determined by Tukey post hoc test. (E) EPAC1 depletion increases the cell population positive for poly(GR)-induced SGs but suppresses the maturation of poly(GR)-induced SGs. SH-SY5Y cells were co-transfected with siRNA and a GFP-$GR_{100}$ expression vector, as described in Fig 6C. The assembly of poly(GR)-induced SGs was quantified as in Fig 1. Data represent means ± SEM ($n$ = 10–12 confocal images obtained from 3 independent experiments; $n$ = 239–275 GFP-$GR_{100}$–positive cells). n.s., not significant; $^{*}P < 0.05$, $^{***}P < 0.001$, as determined by Student $t$ test. All underlying numerical values are available in S1 Data. ANOVA, analysis of variance; EPAC1, exchange protein directly activated by cyclic AMP 1; GFP, green fluorescent protein; ISRIB, integrated stress response inhibitor; LSM12, like-Sm protein 12; NCT, nucleocytoplasmic transport; RAN, repeat-associated non-AUG; S-tdT, S-tdTomato; SEM, standard error of the mean; siRNA, small interfering RNA.
(TIFF)

**S9 Fig. Poly(GR)-induced neurodegeneration is exacerbated in *Drosophila Epac* mutants.**
(A) Loss of *Epac* function exacerbates the neurodegenerative effects of *C9ORF72*-derived poly(GR) proteins in a *Drosophila* eye model. Transgenic $GR_{36}$ protein was overexpressed in photoreceptor neurons (GMR>$GR_{36}$) of wild-type (control), heterozygous *Epac*-deletion mutant (*EpacΔ*), or EPAC-depleted flies (*Epac*^RNAi #1, v50372; *Epac*^RNAi #2, v110077). Representative images of 1-week-old male flies per genotype are shown for ommatidial disorganization and necrotic spots. GMR served as transgenic controls. (B) The eye phenotypes were scored in individual transgenic flies ($n$ = 123–170; weak, moderate, strong), and their relative distribution was calculated for each genotype. $^{***}P < 0.001$, as determined by chi-squared test. (C) Two *Epac* RNAi transgenes (*Epac*^RNAi #1, v50372; *Epac*^RNAi #2, v110077) were individually overexpressed in the *Drosophila* photoreceptor neurons by GMR-GAL4 driver. Total RNA was prepared from fly heads. The abundance of *Epac* transcripts was quantified by real-time RT-PCR and normalized to that of *poly(A)-binding protein*. Relative *Epac* mRNA levels were then calculated by normalizing to those in GMR-GAL4/+ heterozygous controls. Data represent means ± SEM ($n$ = 3). $^{*}P < 0.05$, $^{***}P < 0.001$, as determined by 1-way ANOVA with Dunnett post hoc test. All underlying numerical values are available in S1 Data. ANOVA,

analysis of variance; EPAC, exchange protein directly activated by cyclic AMP 1; GMR, glass multiple reporter; RT-PCR, reverse transcription PCR; SEM, standard error of the mean.
(TIFF)

**S10 Fig. C9-ALS iPSCs and their isogenic control cells exhibit comparable efficiency of motor neuron differentiation.** (A) C9-ALS iPSNs (CS29) and isogenic control iPSNs (CS29-ISO) were fixed 7 days (1-week-old) or 21 days (3-week-old) after neuronal differentiation from their parental NPCs. Immunofluorescence assays were performed using anti-ChAT antibody (red), anti-MAP2 antibody (green), anti-HB9 antibody (magenta), and Hoechst 33258 (blue). The percentages of MAP2-positive (a neuronal marker), ChAT-positive (cholinergic neurons), and HB9-positive cells (motor neurons) were calculated and averaged ($n = 7$ confocal images from 3 independent experiments; $n = 511$–$624$ Hoechst–positive cells). Error bars indicate SEM. n.s., not significant; as determined by 2-way ANOVA with Tukey post hoc test. (B) poly(GR) aggregates are readily detectable in C9-ALS iPSNs, but not isogenic control neurons. iPSNs were co-stained with anti-MAP2 antibody (green), anti-poly(GR) antibody (red), and Hoechst 33258 (blue) similarly as above. All underlying numerical values are available in S1 Data. ANOVA, analysis of variance; C9-ALS, C9ORF72-associated amyotrophic lateral sclerosis; NPC, neural progenitor cell; SEM, standard error of the mean.
(TIFF)

**S11 Fig. Lentiviral overexpression of LSM12 or EPAC1 negligibly affects the efficiency of neuronal differentiation from NPCs.** (A) SH-SY5Y cells were infected with individual recombinant lentiviruses that express the indicated FLAG-tagged proteins along with a GFP reporter. Total cell extracts were prepared 48 hours after infection and immunoblotted with anti-FLAG, anti-GFP, and anti-tubulin (loading control) antibodies. (B) NPCs from C9-ALS iPSCs (CS29) and their isogenic control cells (CS29-ISO) were transduced with the indicated recombinant lentiviruses. Three-week-old iPSNs were fixed and co-stained with anti-MAP2 antibody (magenta) and Hoechst 33258 (blue). (C) The percentages of MAP2–positive neurons among GFP–positive cells were calculated and averaged ($n = 8$ confocal images obtained from 4 independent experiments; $n = 123$–$156$ GFP–positive cells). Error bars indicate SEM. n.s., not significant; as determined by 2-way ANOVA with Tukey post hoc test. All underlying numerical values are available in S1 Data. ANOVA, analysis of variance; EPAC1, exchange protein directly activated by cyclic AMP 1; GFP, green fluorescent protein; iPSC, induced pluripotent stem cell; LSM12, like-Sm protein 12; NPC, neural progenitor cells; SEM, standard error of the mean.
(TIFF)

**S12 Fig. Lentiviral overexpression of LSM12 or EPAC1 suppresses caspase-3 activation in C9-ALS iPSNs.** Three-week-old C9-ALS iPSNs (CS28 and CS52) and control iPSNs (CS0 and CS4) were fixed and co-stained with anti-cleaved caspase-3 antibody, anti-MAP2 antibody, and Hoechst 33258. The relative percentages of iPSNs expressing cleaved caspase-3 were quantified as in Fig 7H. Data represent means ± SEM ($n = 828$–$1,122$ GFP-positive cells from 3 independent differentiation experiments). n.s., not significant; $^*P < 0.05$, $^{**}P < 0.01$, $^{***}P < 0.001$, as determined by 2-way ANOVA with Tukey post hoc test. All underlying numerical values are available in S1 Data. ANOVA, analysis of variance; C9-ALS, C9ORF72-associated amyotrophic lateral sclerosis; EPAC1, exchange protein directly activated by cyclic AMP 1; GFP, green fluorescent protein; LSM12, like-Sm protein 12; SEM, standard error of the mean.
(TIFF)

**S13 Fig. *LSM12* and *EPAC1* promote the association of RAN and Importin β1 with the RANBP2-RANGAP1 complex.** (A) *LSM12* deletion dissociates RAN and importin β1 from the RANBP2-RANGAP1 complex. Soluble extracts from control or *LSM12*$^{KO}$ cells were immunoprecipitated with control IgG or anti-RANGAP1 antibody. IP complexes were analyzed as in Fig 8A. Asterisks indicate SUMOylated RANGAP1. Input, 4.5% of soluble extracts used in each IP. (B) A selective activator of EPAC1 (8-pCPT-2′-O-Me-cAMP-AM/007-AM) does not rescue the association of RAN and importin β1 with the RANBP2-RANGAP1 complex in *LSM12*-deleted cells. Where indicated, cells were preincubated with 1-μM 007-AM or DMSO (vehicle control) at 37°C for 1 hour before immunoprecipitating soluble cell extracts with control IgG or anti-RANBP2 antibody. (C, D) EPAC1 depletion dissociates RAN and importin β1 from the RANBP2-RANGAP1 complex. Soluble extracts from control or *EPAC1* siRNA-transfected cells were immunoprecipitated with anti-RANBP2 (C) or anti-RANGAP1 antibodies (D). EPAC1, exchange protein directly activated by cyclic AMP 1; IP, immunoprecipitation; LSM12, like-Sm protein 12; RAN, repeat-associated non-AUG; RANBP2, Ran-binding protein 2; RANGAP1, RAN GTPase-activating protein 1.
(TIFF)

**S14 Fig. RAN overexpression suppresses the effects of EPAC1 depletion on the poly(GR)-induced disruption of NCT.** (A) SH-SY5Y cells were co-transfected with siRNA and expression vectors for S-tdT, GFP-GR$_{100}$, and RAN-FLAG, as in Fig 6A. Transfected cells were co-stained with anti-FLAG antibody (magenta) and Hoechst 33258 (blue) 48 hours after plasmid DNA transfection. (B) NCT of S-tdT reporter proteins was quantified as in Fig 2A. Two-way ANOVA detected significant interaction effects of GFP-GR$_{100}$ and EPAC1 depletion on NCT only in FLAG-expressing cells ($P = 0.0005$ for *EPAC1*$^{siRNA}$ #1; $P = 0.0093$ for *EPAC1*$^{siRNA}$ #2). Data represent means ± SEM ($n = 113–115$ GFP–or GFP-GR$_{100}$–positive cells from 3 independent experiments). n.s., not significant; $^{*}P < 0.05$, $^{**}P < 0.01$, $^{***}P < 0.001$, as determined by Tukey post hoc test. All underlying numerical values are available in S1 Data. GFP, green fluorescent protein; NCT, nucleocytoplasmic transport; RAN, repeat-associated non-AUG; S-tdT, S-tdTomato; SEM, standard error of the mean; siRNA, small interfering RNA.
(TIFF)

**S15 Fig. A model for the *LSM12-EPAC1* pathway that establishes the nucleocytoplasmic RAN gradient and suppresses the toxicity of *C9ORF72*-derived poly(GR) protein.** The *LSM12-EPAC1* pathway contributes to the assembly of the RAN-associating protein complex at the cytoplasmic side of the nuclear pore, thereby facilitating RAN-GTP recycling. The loss-of-function of the *LSM12-EPAC1* pathway dissociates RAN and importin β1 from the RANBP2-RANGAP1 complex and delays the nuclear entry of RAN for recycling. Disruption of the RAN gradient impairs NCT, promotes the nuclear assembly of poly(GR) granules, and increases poly(GR) toxicity (e.g., loss of nuclear laminar integrity and degeneration of *Drosophila* photoreceptor neurons). Asterisks indicate stronger interaction between RANBP2 and RANGAP1 caused by the loss of the *LSM12-EPAC1* pathway. All underlying numerical values are available in S1 Data. EPAC1, exchange protein directly activated by cyclic AMP 1; LSM12, like-Sm protein 12; RAN, repeat-associated non-AUG; RANBP2, Ran-binding protein 2; RANGAP1, RAN GTPase-activating protein 1.
(TIFF)

**S1 Data. Numerical raw data in Figs 1–8 and S1–S12 and S14 Figs.**
(XLSX)

**S1 Raw Images. Original blot data in Figs 1, 5, 7 and 8 and S2, S5, S7, S8, S11 and S13 Figs.**
(PDF)

## Acknowledgments

We thank X. Cheng, A. Gitler, M.S. Hipp, D. Ito, H.M. Kwon, C.Y. Park, J.D. Rothstein, B. Weinberg, Addgene, Bloomington *Drosophila* Stock Center, Vienna *Drosophila* Resource Center, and Developmental Studies Hybridoma Bank for reagents.

## Author Contributions

**Conceptualization:** Chunghun Lim.

**Formal analysis:** Jongbo Lee, Chunghun Lim.

**Funding acquisition:** Ki-Jun Yoon, Yoon Ki Kim, Chunghun Lim.

**Investigation:** Jongbo Lee.

**Methodology:** Jongbo Lee, Jumin Park, Ji-hyung Kim, Giwook Lee, Tae-Eun Park, Ki-Jun Yoon, Yoon Ki Kim, Chunghun Lim.

**Supervision:** Chunghun Lim.

**Visualization:** Jongbo Lee, Chunghun Lim.

**Writing – original draft:** Jongbo Lee.

**Writing – review & editing:** Jongbo Lee, Chunghun Lim.

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
