## [Editor Report · Decision Letter 0]

10 Jun 2020

Dear Dr Lim, 

Thank you for submitting the revision of your manuscript entitled "LSM12 -EPAC1 Defines a Neuroprotective Pathway that Sustains the Nucleocytoplasmic RAN Gradient" for consideration as a Research Article by PLOS Biology.

I have checked the revision and consulted with the Academic Editor, and I am writing to let you know that we would like to send your submission out for external peer review. As this is a new submission, we need you to complete your submission by providing again the metadata that is required for full assessment. To this end, please login to Editorial Manager where you will find the paper in the 'Submissions Needing Revisions' folder on your homepage. Please click 'Revise Submission' from the Action Links and complete all additional questions in the submission questionnaire.

Please re-submit your manuscript within two working days, i.e. by Jun 12 2020 11:59PM.

Kind regards,

Ines

--

Ines Alvarez-Garcia, PhD

Senior Editor

PLOS Biology

Carlyle House, Carlyle Road

Cambridge, CB4 3DN

---

## [Decision Letter · Decision Letter 1]

3 Aug 2020

Dear Dr Lim,

Thank you very much for submitting a revised version of your manuscript "LSM12-EPAC1 Defines a Neuroprotective Pathway that Sustains the Nucleocytoplasmic RAN Gradient" for consideration as a Research Article at PLOS Biology. This revised version of your manuscript has been evaluated by the PLOS Biology editors, the Academic Editor and the three original reviewers.

You will see that the reviewers think the manuscript is very much improved and both Reviewers 1 and 3 are mostly satisfied. Reviewer 2, however, still raises two concerns that should be addressed before we can consider the manuscript for publication. One is the fact that a single C9 iPS cell line is used in the experiments and that this is below the standards in the field, thus we deem essential that more than one iPSC line is used for the experiments. In addition, you will need to show that LSM12-EPAC1 rescue neuronal death or neurodegenerative phenotypes, as this reviewer requests.

In light of the reviews (attached below), we will not be able to accept the current version of the manuscript, but we would welcome re-submission of a much-revised version that takes into account the reviewers' comments. We cannot make any decision about publication until we have seen the revised manuscript and your response to the reviewers' comments. Your revised manuscript is also likely to be sent for further evaluation by the reviewers.

We expect to receive your revised manuscript within 3 months. 

**IMPORTANT - SUBMITTING YOUR REVISION**

*Re-submission Checklist*

*Published Peer Review*

*PLOS Data Policy*

*Blot and Gel Data Policy*

Sincerely,

Ines

--

Ines Alvarez-Garcia, PhD,

Senior Editor,

ialvarez-garcia@plos.org,

PLOS Biology

Reviewers’ comments

Rev. 1:

The authors have addressed all my concerns. I am happy with the manuscript in its form.

Rev. 2:

This manuscript from Lee and colleagues is much improved from the initial submission. Unfortunately, though, the new and potentially exciting data in iPS neurons does not meet minimum standards of rigor for publication. Only a single C9 iPS cell line is used, far less than the typical 5 line minimum considered standard in the field. Furthermore, there is no data in this cell line characterizing the number of G4C2 repeats in the C9-HRE line or in the corrected c9orf72 gene (how many G4C2 repeats in the isogenic control?). To my knowledge, this line has not been published, even though it is commercially available from Cedars-Sinai. For example, see Coyne et al, Neuron, 2020 which used 20 iPSC cell lines, including one isogenic pair, but did not use this isogenic pair. Additional cell lines are critical given the new data suggesting that LSM12 and EPAC1 expression are reduced, suggesting for the first time in this paper, that these genes may play a role in C9 pathogenesis. If these findings were confirmed in multiple cell lines, this would greatly increase this reviewer's enthusiasm for publication.

A second major concern I still have is the lack of evidence to support the title of the paper "LSM12-EPAC1 Defines a Neuroprotective Pathway that Sustains the Nucleocytoplasmic RAN Gradient". All three reviewers of the initial submission asked the authors for a relatively simple experiment to test their hypothesis: overexpress LSM12 and/or EPAC1 in their Drosophila model of C9-ALS, but the results of this experiment has not been described by the authors - presumably they were unable to perform the experiment or it gave the opposite results that they would have predicted. Instead, they overexpressed these genes in cell lines and show that it restores the Ran gradient and TDP-43 mislocalization. However, this is NOT sufficient evidence to claim that LSM12-EPAC1 are neuroprotective - this would require showing that LSM12-EPAC1 rescue neuronal death or neurodegenerative phenotypes, neither of which are shown in this paper.

Rev. 3:

The authors have submitted a substantially revised manuscript and most of my comments have been answered. The addition of human patient-derived neurons has much improved the manuscript and in its current form the manuscript is suitable for publication in PLos Biology.

I have few minor comments, that would need to be addressed in the manuscript:

1) How long were the patient-derived neurons in culture and when were the experiments performed, DIV? would be good to include in the figure.

2) What was the infection efficiency and were any survival measurements done?

3) Please provide some low magnification images showing large field of view for infection of neurons in Supplementary figure

4) Was any Filter trap assays performed on patient-derived neurons to measure soluble versus insoluble Poly-GR aggregates

The manuscript has a found a novel and interesting pathway that serves as a disease pathway and these findings are of interest to the field.

---

## [Editor Report · Decision Letter 2]

22 Oct 2020

Dear Dr Lim,

Thank you for submitting the new revised version of your Research Article entitled "LSM12-EPAC1 Defines a Neuroprotective Pathway that Sustains the Nucleocytoplasmic RAN Gradient" for publication in PLOS Biology. I have now obtained advice from the Academic Editor and discussed the revision with the team of editors. 

We're delighted to let you know that we're now editorially satisfied with your manuscript. However before we can formally accept your paper and consider it "in press", we also need to ensure that your article conforms to our guidelines. A member of our team will be in touch shortly with a set of requests. As we can't proceed until these requirements are met, your swift response will help prevent delays to publication. Please also make sure to address the data and other policy-related requests noted at the end of this email.

- a cover letter that should detail your responses to any editorial requests, if applicable

*Copyediting*

*Published Peer Review History*

*Early Version*

Sincerely,

Ines

--

Ines Alvarez-Garcia, PhD,

Senior Editor,

ialvarez-garcia@plos.org,

PLOS Biology

Fig. 1B-E; Fig. 2B, D; Fig. 3B, D, F, G; Fig. 4B, D, F; Fig. 5A, B, C, D, F, H; Fig. 6B, D, F, H, J; Fig. 7A, B, D, F, H; Fig. 8D, F; Fig. S1A, C; Fig. S2B, C; Fig. S3A, B; Fig. S4B, C; Fig. S5B; Fig. S6; Fig. S7A, C; Fig. S8B, D, E; Fig. S9B, C; Fig. S10A; Fig. S11C; Fig. S12 and Fig. S14B

We note that you were planing to deposit the RNA and TRAP sequencing data in GEO, so please do so and let us know the GEO number.

---

## [Editor Report · Decision Letter 3]

19 Nov 2020

Dear Dr Lim,

On behalf of my colleagues and the Academic Editor, Gillian P Bates, I am pleased to inform you that we will be delighted to publish your Research Article in PLOS Biology. 

PRODUCTION PROCESS

Before publication you will see the copyedited word document (within 5 business days) and a PDF proof shortly after that. The copyeditor will be in touch shortly before sending you the copyedited Word document. We will make some revisions at copyediting stage to conform to our general style, and for clarification. When you receive this version you should check and revise it very carefully, including figures, tables, references, and supporting information, because corrections at the next stage (proofs) will be strictly limited to (1) errors in author names or affiliations, (2) errors of scientific fact that would cause misunderstandings to readers, and (3) printer's (introduced) errors. Please return the copyedited file within 2 business days in order to ensure timely delivery of the PDF proof. 

If you are likely to be away when either this document or the proof is sent, please ensure we have contact information of a second person, as we will need you to respond quickly at each point. Given the disruptions resulting from the ongoing COVID-19 pandemic, there may be delays in the production process. We apologise in advance for any inconvenience caused and will do our best to minimize impact as far as possible.

EARLY VERSION

PRESS 

Kind regards,

Erin O'Loughlin

Publishing Editor, 

PLOS Biology

on behalf of

Ines Alvarez-Garcia,

Senior Editor

PLOS Biology